

# Ionospheric Plasma Irregularities During Intense geomagnetic storms of Solar Cycle 25

Nadia Imtiaz[1], Andres Calabia[2, 3], Chukwuma Anoruo[4], Aqsa Zahid[5], Christine Amory[6], and Binod Adhikari[7]

[1]Theoretical Physics Division, PINSTECH, Pakistan
[2]School of Surveying and Land Information Engineering, Henan Polytechnic University, Jiaozuo, China.
[3]Department of Physics and Mathematics, University of Alcalá, 28801-Alcalá de Henares (Madrid), Spain.
[4]Laboratório de Física e Astronomia, Universidade do Vale do Paraíba, Av. Shishima Hifumi, 2911–Urbanova, São José Dos Campos (SP), Brazil
[5]Department of Physics, Quaid-i-Azam University, Pakistan
[6]Plasmas (LPP), CNRS, Sorbonne Université, Université Paris Saclay, Observatoire de Paris, Ecole polytechnique, Institut Polytechnique de Paris, Paris, France, 3
[7]Department of Physics, Patan M. Campus, Tribhuvan University, Lalitpur, Nepal

**Correspondence:** Nadia Imtiaz (nhussain@ualberta.ca)

**Abstract.** This study aims to characterize several key aspects of the ionosphere during intense geomagnetic storms that occurred on March 23-25, 2023, April 23-25, 2023, November 4-7, 2023 and May 10-13, 2024 during the ascending phase of Solar Cycle 25 (SC25). We are especially interested in studying the role of asymmetric Joule Heating (JH) in the structuring of the Equatorial Ionization Anomaly (EIA), such as double crest, single crest, or merged, which may lead to the formation

or suppression of post-sunset ionospheric plasma irregularities. For this purpose, we used the Weimer 2005 model simulations to analyze the JH patterns during the four strong geomagnetic storms, and Madrigal TEC maps are used to observe changes in the intensity, location, and symmetry of the EIA during these disruptive times. Equatorial/low-latitude ionospheric plasma irregularities at different longitudes under geomagnetically disturbed conditions are studied using the Rate of Change TEC Index (ROTI), which is calculated from GPS receiver measurements. A strong JH is observed during the May 2024 storm

(also known as the Mother's Day storm) during its main phase, which occurs after sunset between 18:00 and 00:00 UT. The other storms have JH strength in the following order from strong to weak: March 2023, April 2023, and November 2023. In addition to interhemispheric asymmetry, all the storms show stronger variation in the JH patterns. We conclude that the resulting change in the thermospheric winds and electric fields due to storm conditions alters the EIA structures. It has been found that the generation of ionospheric plasma irregularities and their geographical distribution strongly depend on EIA's density

gradients and general structure. For instance, it is noticed that the double crest EIA structures with strong plasma density gradients play an important role in the generation of post-sunset ionospheric plasma irregularities during the main phases of these geomagnetic storms. However, the single-crest or merged EIA structure comprises a diffuse region of high electron density centered directly over the equator, without a pronounced trough, as observed during the storm of November 2023. The single crest EIA exhibit nearly uniform plasma density distribution, do not favor the generation of ionospheric plasma irregularities.

The role of a storm-time penetrating electric field in the structuring and seeding of ionospheric plasma irregularities has been





investigated. The research will contribute to our understanding of the basic physics of the ionosphere, especially the mechanisms governing the development and evolution of the EIA and ionospheric plasma irregularities under various magnetically disturbed conditions.

## 1 INTRODUCTION

The study of ionospheric properties, particularly the equatorial ionosphere, is critical because of its influence on radio signal transmission, such as those from Global Navigation Satellite Systems (GNSS). The ionosphere is an ionized region of the upper atmosphere that ranges from 60 to 1000 kilometers above the Earth's surface and is separated into three major regions: D (50-90 km), E (90-150 km), and F (150-500 km). These regions with lack of a clear boundaries exhibit diurnal and seasonal fluctuations, especially during magnetic storms, when changes in both thermosphere and ionospheric parameters such

as neutral wind, neutral composition, and ionospheric electric field deviate from their regular patterns, describing upper atmosphere variability (Balan et al., 2018), and generating significant disturbances in GNSS (Zhang et al., 2020). Ionospheric effects on applications are generally minor in mid-latitudes, causing only small perturbations in signal propagation that may be roughly expected, especially during magnetic storms. However, the equatorial region experiences more severe disturbances and irregular ionospheric structures. Storm-related variations in thermosphere/ionosphere parameters can dramatically alter the

low-latitude ionosphere phenomena, notably the equatorial plasma fountain (EPF), equatorial ionization anomaly (EIA) and ionospheric plasma irregularities (Kelly, 2009; Kassa et al., 2015). The EPF consists of two components: the upward drift force $E \times B$, which lifts the plasma to higher altitudes away from the magnetic equator, and the field-aligned diffusion of plasma along the magnetic field lines at all altitudes. The EIA is a feature of the non-uniform latitudinal distribution of ionospheric electron density caused by the upward $E \times B$ drift, with two crests at magnetic latitudes of about $15°$ north/south and a trough

near the magnetic dip equator (Balan et al., 2018). The EPF, the EIA, and the density gradients in the EIA would all be affected by the zonal electric fields. EPF variations may occur due to zonal electric fields during magnetic storms, which may also increase daytime eastward electric fields.

Storm-time electric fields can emerge from either a short-lived prompt penetration electric field (PPEF) caused by solar wind-magnetospheric interaction or a disrupted dynamo (DD) electric field. The southward turning of the IMF $B_z$ component causes

a rapid increase in high-latitude convection and substorm activity. Convective electric fields at high latitudes can expand much faster than the magnetosphere can respond, which enables them to penetrate the low-latitude ionosphere. The PPEF polarity is eastward during the day and westward during the night. Additionally, the region 2 current builds up the shielding layer due to the enhanced convection, allowing an overshielding electric field opposite to the PPEF polarity to penetrate the low-latitude ionosphere when the IMF $B_z$ turns northward. This penetration can be as efficient as when the IMF $B_z$ turns South. The

DD electric field is a manifestation of disturbed thermospheric winds. Storm-time high-latitude heating enhances meridional neutral winds, resulting in strong equatorial winds. At low latitudes, these disturbance winds generate a westward/eastward electric field on the dayside and nightside, contradicting their quiet-time tendencies. Compared to the PPEF, the disturbance dynamo forms more slowly. Because of neutral air inertia, the disturbance winds and related dynamo electric fields take many



hours to form and may last for several hours (Astafyeva et al., 2018; Blanc and Richmond, 1980; Huang et al., 2005; Kikuchi
et al., 2008). Moreover, the equatorial vertical plasma drifts associated with the PPEF and DD exhibit seasonal and longitudinal
dependence (Fejer et al., 1999).

Aside from storm-time electric fields, the disturbed thermospheric equator-ward neutral wind can carry plasma from one hemi-
sphere to the other and, depending on its amplitude and direction, alter the low-latitude ionosphere. Storm-time continuous
energy input in the high latitude ionosphere-thermosphere coupling system causes the neutral atmosphere to rapidly expand.
This growth can produce variations on the global distribution of the O/N2 ratio. Such disturbances in the neutral composition
also influence the EIA, either positively by increasing plasma density or negatively by reducing plasma density, depending on
the relative shift in O to N2 ratio distribution (Li et al., 2018; Rishbeth, 2000; Kassa et al., 2015). Thermosphere circulation
impacts the relative strength of EIA crests in the summer and winter hemispheres, leading to the EIA hemispheric asymmetry
(Huang et al., 2005; Tulasi et al., 2009). The trans-equatorial neutral wind from the summer to the winter hemisphere raises
electron density at the winter hemisphere's crest. Also, an increase in solar extreme ultraviolet (EUV) in January caused by a
decrease in the Sun-Earth distance at perihelion resulted in a $20\%$ greater electron density in the northern EIA crest than in
the southern EIA crest (Lin et al., 2005). As mentioned above, the EIA is defined by two plasma density crests around $15°$
from the magnetic equator, is an important phenomenon in the ionosphere F region at low latitudes (Appleton, 1946; Balan
et al., 2018; Wu et al., 2023). The "fountain effect" created by vertical plasma drift and diffusion usually explains the formation
of EIAs. When combined with ambipolar diffusion, an eastward daytime dynamo electric field at the dip equator generates
an upward drift that lifts the ionospheric plasma to higher altitudes, resulting in two plasma density crests at $15°$ magnetic
latitudes in the ionospheric F region. Storm-induced perturbations in electric fields, neutral winds, and neutral composition all
affect EIA evolution during storms. The storm-time expansion in the EIA to mid-latitudes and considerable density changes are
caused by eastward PPEF and disturbed meridional winds (Balan et al., 2009; Tsurutani et al., 2004). Aside from double-crest
EIA, single-crest or merged EIA over or near the magnetic equator can also develop under certain ionospheric conditions. The
stronger (weaker) fountain effect, is responsible for the stronger (weaker) and extended (contracted) EIA crests. At low and
equatorial latitudes, a single crest may be observed if the two crests are suppressed by a weaker fountain effect. Huang et al.
(2013) studied that the single crest in the TEC over $110°E$ longitude was primarily happened during periods of extremely
low solar activity. Balan et al. (2013) showed that the merging of the daytime EIA crests during the recovery phase (RP) of
several storms can be attributed to plasma convergence over the equator caused by equatorward storm-time neutral winds, with
minor contributions from downward $\boldsymbol{E} \times \boldsymbol{B}$ drift and an increase in O/N2 ratios. Here, we employ Madrigal TEC database to
examine the formation of single and double crest EIA structures under different physical conditions that occurred during the
ascending phase of the SC25.

Plasma depletions (or irregularities) are significant localized decreases in ionospheric F-region plasma density induced by the
development of a Rayleigh-Taylor (R-T) instability after sunset, which causes severe radio signal disruptions while passing
through them. The majority of plasma depletions occur in the EIA region (M et al., 2012). Vaggu et al. (2023) studied the
morphological and spectral aspects of ionospheric irregularities at E and F regions over Poker Flat using modeling and obser-
vations. Their analysis showed that the E-region irregularities extend along the magnetic field lines, with rod-shaped structures,





whereas the F-region irregularities have wing-like structures, with irregularities extending both along and across the magnetic
fields. It was also discovered that the spectral index or the power-law index of the E-region event is lower than that of the
F-region events. Furthermore, the spectral slope on the ground at higher frequencies is smaller than the spectral slope at irreg-
ularity height.

Previous studies have shown that geomagnetic storms of similar strength can frequently produce different ionospheric re-
sponse. For instance, Aa et al. (2020) presented a statistical analysis of EPIs utilizing in situ plasma density observations
from the Swarm constellation from December 2013 to December 2019. The occurrence patterns of both post-sunset and post-
midnight EPIs were explored in terms of longitude, season, local time, latitude, solar activity, and geomagnetic activity level.
The main results were as follows: (1) The occurrence frequencies of post-sunset/post-midnight EPIs showed varying longitu-
dinal and seasonal dependence: The post-sunset EPIs showed the highest occurrence rate over the American-Atlantic sectors
during the December solstices and equinoxes, whereas the post-midnight EPIs showed the highest occurrence rate during the
June solstices, particularly over the African sector. (2) Post-sunset EPI occurrence rates were positively correlated with solar
activity, but post-midnight EPIs were adversely correlated. (3) The latitudinal distribution of EPIs showed a double-peak pat-
tern about $\pm 5°$ magnetic latitude, with a higher peak in summer. (4) As geomagnetic activity increases, so does the occurrence
of EPIs. (5) The magnetic declination angle, equatorial vertical $\boldsymbol{E} \times \boldsymbol{B}$ drift, and thermospheric zonal wind are the primary
factors influencing the distribution of EPIs after sunset. The key governing factors for post-midnight EPIs are thought to be
atmospheric gravity waves and equatorially directed thermospheric meridional wind associated with the midnight temperature
maxima.

The complex interaction of the magnetosphere, ionosphere, and thermosphere causes variations in ionospheric disturbances
from storm to storm. These disturbances are more noticeable and frequent at night, when high-latitude energy injection occurs
and neutral winds favor an equatorial direction (Fuller-Rowell et al., 1997). In low-latitude and equatorial regions, intense
geomagnetic storms can create ionospheric plasma irregularities, which are most noticeable at night. Storm-time uplift of the
ionospheric F-layer increases the likelihood of RT instability-induced plasma irregularities near the magnetic equator after
sunset. These irregularities can be as small as a few centimeters or as large as thousands of kilometers (Abdu et al., 2009).
The vertical $\boldsymbol{E} \times \boldsymbol{B}$ plasma drift and the strong plasma density gradient near equator can cause gravitational RT-instability
in the nighttime ionosphere, leading to post-sunset plasma irregularities at equatorial/low latitudes (González, 2022; Calabia
et al., 2024; Imtiaz et al., 2024; Tahir et al., 2024; Kassa et al., 2024). Eastward (westward) storm time electric fields can cre-
ate favorable (unfavorable) conditions for irregularity generation (suppression) by raising (lowering) the F-region ionosphere
near the magnetic equator. Pre-reversal enhancement (PRE) in the $\boldsymbol{E} \times \boldsymbol{B}$ drift can cause an uplift of the F-layer which can
provide ideal conditions for the formation of irregularities via RT-instability. Studies showed that the strong storm-time drivers
can increase or decrease the vertical $\boldsymbol{E} \times \boldsymbol{B}$ drift that may initiate or suppress the ionospheric plasma irregularities during a
storm (Aarons, 1991, 1997; Tulasi et al., 2009). The vertical transport of plasma, which is primarily responsible for irregularity
formation, occurs due to the effect of a zonal electric field generated by dynamo action inside the ionosphere or an electric field
imposed from the magnetosphere (Abdu, 2020). The suppression of ionospheric plasma irregularities could be attributed to a
westward PPEF of magnetospheric origin caused by northward IMF B$_z$, as well as DD caused by storm-induced equatorial





winds (Huang et al., 2013; Kassa et al., 2023; Imtiaz et al., 2024; Kassa et al., 2024). In this context, Huang (2019) reported

that the presence of the long term eastward penetration electric field in the equatorial ionosphere, enhanced vertical ion drift in the upward direction by up to 180 m/s. Wu et al. (2023) used MAGE simulations and ICON IVM observations to show that eastward/westward electric field lead to upward/downward ion drifts during the geomagnetic storm of November 3-4, 2023. Multiple factors, including solar and geomagnetic activity, latitude and longitude, local time, and season, influence ionospheric irregularities (Abdu, 1997; Dugassa et al., 2019). To better understand their variability, it is necessary to analyze the occurrence

characteristics of ionospheric plasma irregularities under different geomagnetic conditions.

In this study, we investigate different storm-time ionospheric conditions that characterize the structural forms of the EIA, leading to post-sunset plasma irregularities. For this purpose, we use madrigal TEC maps to study the EIA features such as separated or merged crests after sunset during the main phases of the intense geomagnetic storms of the 25th solar cycle. The GNSS based ROTI is used to study the post-sunset ionospheric irregularities during magnetically disturbed conditions.

In addition, PPEF observations are used to investigate the role of low-latitude ionospheric electric field in the formation of different structures of the EIA which are favorable for R-T instability to generate ionospheric plasma irregularities. The article is organized as follows: Section 2 describes the data and analysis approach, whereas Section 3 presents the results and their interpretations. Finally, Section 4 presents our findings and conclusions.

## 2   DATASETS and MODELS

We use 1-minute and hourly time-resolution data of solar wind parameters such as the $B_z$ component of the interplanetary magnetic field (IMF), the solar wind speed ($V_{sw}$), the eastward zonal component $E_y$ of the interplanetary electric field (IEF), the ring current proxy index ($SYM - H$), the geomagnetic activity index (Kp), and the solar radio flux at 10.7 cm (F10.7). The data can be found at https://omniweb.gsfc.nasa.gov/form/omni_min.html in the NASA OMNIWeb database.

The Community Coordinated Modeling Center (CCMC) hosted Weimer-2005 model driven by ACE solar wind data is used to

study the high-latitude ionospheric electrodynamics (Weimer, 2005). It is an improved empirical model to compute magnetic field-aligned currents (FACs) and high-latitude electric potentials as a function of solar wind properties. As an analog of the electric potential model, the FAC model is constructed using scalar magnetic Euler potentials. The potentials are constructed from several Fourier series functions of longitude at discrete latitudinal intervals at lower latitudes. Joule heating (JH) and total Poynting flux in the ionosphere can be calculated using the FAC and electric potential models. Since the FAC model implicitly

incorporates the conductivity variations as a function of solar tilt, ionospheric currents and JH can be derived without the need for a separate model of ionospheric conductivity.

Worldwide TEC maps, updated every 5 minutes, are available in the CEDAR Madrigal database. The maps have a spatial resolution of $1° \times 1°$ for geographic latitude (GLAT) and longitude (GLON). The data is sourced from ground-based GNSS signals, which are publicly available at http://cedar.openmadrigal.org/. Rideout and Coster (2006) provides a comprehensive

understanding of data processing techniques of these maps.

The Rate of Total Electron Content Index (ROTI), which interprets the TEC's variability over time, is an indicator of iono-



**Table 1.** Information about the location and local time (LT) of GNSS stations used in the analysis.

| Instrument | Station | Sector | Geographic (Latitude, Longitude) | Geomagnetic (Latitude, Longitude) | Local Time (LT) |
|---|---|---|---|---|---|
| GNSS | BAKO | Asia | $(6.49°S, 106.85°E)$ | $(16.13°S, 179.93°W)$ | UT+7 |
| – | DGAR | – | $(6.49°S, 72.37°E)$ | $(14.94°S, 143.93°W)$ | UT+5 |
| – | IISC | – | $(13.02°N, 77.57°E)$ | $(4.87°N, 150.96°W)$ | UT+5 |
| – | CUSV | – | $(13.74°N, 100.53°E)$ | $(4.39°N, 173.37°W)$ | UT+7 |
| – | HKSL | – | $(22.37°N, 113.93°E)$ | $(13.19°N, 173.15°W)$ | UT+7 |
| – | RABT | Africa | $(34.1°N, 6.50°W)$ | $(23.88°N, 69.23°E)$ | UT+0 |
| – | YKRO | – | $(6.86°S, 5.24°W)$ | $(10.92°N, 68.67°E)$ | UT+0 |
| – | ABPO | – | $(19.02°S, 47.23°E)$ | $(23.49°S, 116.71°E)$ | UT+3 |
| – | BOGT | America | $(4.64°N, 74.08°W)$ | $(14.31°N, 1.46°W)$ | UT-5 |
| – | RIOP | – | $(1.65°S, 78.65°W)$ | $(7.23°N, 6.46°W)$ | UT-5 |
| – | CORD | – | $(31.53°S, 64.47°W)$ | $(22.43°S, 7.27°E)$ | UT-4 |
| – | SANT | – | $(33.15°S, 70.67°W)$ | $(23.77°S, 1.83°E)$ | UT-5 |

spheric irregularities (Aarons, 1997; Basu et al., 1999; Pi et al., 1997). GNSS-TEC software by (Seemala, 2011) solves for receiver biases and estimates the TEC from satellites with elevation angles $\geq 30°$ to minimize multi-path errors. Given TEC data at discrete intervals, the rate of change of TEC (ROT) can be computed as:

$$\text{ROT}(t_i) = \frac{d\text{TEC}}{dt} = \frac{TEC(t_{i+1}) - TEC(t_i)}{t_{i+1} - t_i}. \tag{1}$$

In this equation, $d\text{TEC}$ is the difference between $\text{TEC}(t_{i+1})$ and $\text{TEC}(t_i)$ values at times $t_{i+1}$ and $t_i$, respectively. ROTI is then calculated as the standard deviation of ROT over a specified period, usually 5 minutes,

$$\text{ROTI} = \sqrt{<\text{ROT}^2> - <\text{ROT}>^2}, \tag{2}$$

Table 1 shows the geographic and geomagnetic coordinates of equatorial and low-latitude GNSS receivers in the Asian, African,
and American sectors, which are used to examine the temporal patterns of ROTI. For the categorization of ionospheric irregularities, we used ROTI values as follows: a ROTI value of less than 0.25 TECU/min indicates the absence of irregularities; a weak irregularity is identified when ROTI ranges from 0.25 to 0.5 TECU/min; a moderate irregularity is noted when ROTI falls between 0.5 and 1 TECU/min; and a strong irregularity is present when ROTI exceeds 1 TECU/min.

The temporal profile of the equatorial electric field over a particular longitude is obtained from the real-time Prompt Penetration
Electric Field (PPEF) model, available at: https://geomag.colorado.edu/online-calculators/real-time-model-ionospheric-electric-fields. Employing real-time solar wind data and a climatological model for quiet conditions, the model calculates the temporal variation of the equatorial ionospheric eastward electric field (EEF) using a transfer function. Time and longitude are the inputs, and the outcomes are the total electric field (QP), i.e., the sum of background quiet-time electric field (Q) and the PPEF.



**Table 2.** Information about the Sudden Storm Commencement (SSC), the main phase end (MPE), the minimum values of the SYM-H index, the number of Sun spots (R), and Quiet days (Q) during different geomagnetic storms of the solar cycle 25.

| Event | SSC | MPE | $SYM - H_{min}$ | R | Q days |
|---|---|---|---|---|---|
| March 23-25, 2023 | March 23 at 6:32 UT | March 24 at 5:21 UT | -170 nT | 150 | March 17-19 |
| April 23-25, 2023 | April 23 at 8:56 UT | April 24 at 4:03 UT | -233 nT | 80 | April 20-22 |
| November 4-6, 2023 | November 4 at 17:40 UT | November 5 at 16:54 UT | -189 nT | 91 | November 1-3 |
| May 10-13, 2024 | May 10 at 17:15 UT | May 11 at 02:14 UT | -518 nT | 200 | May 7-9 |

The information given in this table is obtained from OMNI web database.

# 3  RESULTS AND DISCUSSION

## 3.1  SOLAR AND MAGNETOSPHERE CONDITIONS

The investigation of the ionospheric response to geomagnetic storms reveals insight into the complex interplay between solar activity and Earth's atmosphere. In order to achieve this, we investigate four geomagnetic storms that occurred during the ascending phase of Solar Cycle 25. Figures 1-4 show (a) the $B_z$ component of the IMF, and the solar wind speed ($V_{sw}$), (b) the zonal component eastward $E_y$ of the IEF, and the ring current proxy index ($SYM - H$), and (c) the geomagnetic activity index (Kp), and the solar radio flux at 10.7 cm (F10.7) during the storm events considered. Information about the sudden storm commencement (SSC), the end of the main phase (MP), the minimum values of the SYM-H index, the number of Sun spots (R) and the magnetic quiet days (Q) during different geomagnetic storms is also given in Table 2. Following is a brief overview of the geomagnetic conditions during these events:

**March 23-25, 2023 storm:** As shown in Figure 1a, the IMF $B_z$ makes two large southward turns throughout the main phase of the storm. The first southward turn occurs on March 23, at approximately 6:32 UT. Between 6:32 and 12:00 UT, the value of IMF $B_z$ decreased from 2 to -10 nT. The second long-term southward turn in IMF $B_z$ is seen on March 23, at 17:22 UT. During this turn, the IMF $B_z$ value decreased from 10 nT to approximately -18 nT. The IMF $B_z$ remains southward for approximately 9 hours before increasing around 5:21 UT on March 24. During the main phase of this storm, the $V_{sw}$ fluctuated between 400 and 500 km/s. During the recovery phase of this storm, the $V_{sw}$ increased and reached the maximum value of 600 km/s on March 25. A northward IMF $B_z$ generates an electric field westward during the day and eastward at night. Figure 1 b shows that the IEF $E_y$ reached about 4 mV / m after the first southward excursion of the IMF $B_z$. From 12:00 to 18:00 UT, on March 23, it fluctuated rapidly between +5 and -5 mV/m. During the second southward excursion, it again increases and reached the maximum value of 8.22 mV/m at 17:22 on March 23. After that, it started decreasing down to normal conditions. Figure 1c shows that the main phase of this storm started soon after SYM-H reached a value of 28 nT at the time of SSC. During the extended southward turning of IMF $B_z$ ($B_z < 0$), enhanced ring current caused by magnetic reconnection dropped the SYM-H index to -170 nT on March 24, at 5:21 UT (end of the main phase). The SYM-H increases during the recovery phase until it reached the quiet time value. Figure 1 c shows that before SSC, Kp was less than 4+. On March 24, around 04:00 UT, the Kp value peaked at 8+ and then began to decrease as the storm recovery began around 5:21 UT. The F10.7 index





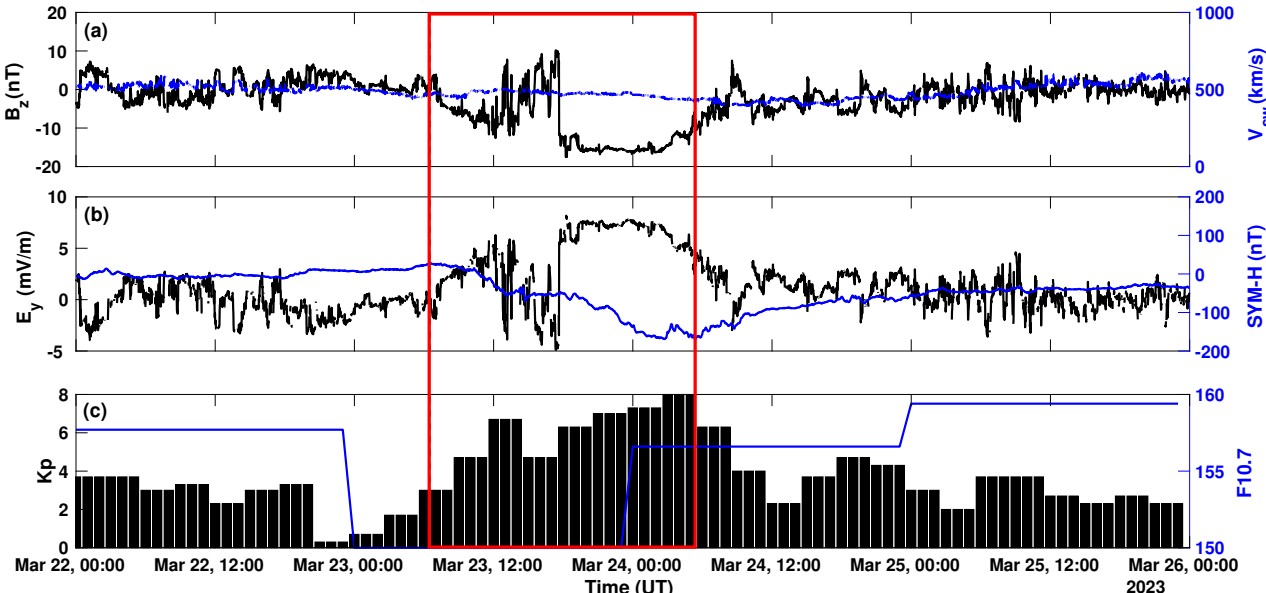

**Figure 1.** Temporal variation of: (a) solar wind speed ($V_{sw}$) and $B_z$ component of IMF, (b) $E_y$ component of IMF and $SYM - H$, and (c) the magnetic index ($K_P$) and the solar radio flux at 10.7 cm (F10.7) during March 22-25, 2023 storm.

fluctuated between 157.7 and 159.4 sfu during this period. As a result, variations in solar irradiance are expected to have little

impact on thermospheric composition during the storm.

**April 23-25, 2023 storm:** Figure 2(a-c) shows the solar wind and geomagnetic conditions during the period April 22-26, 2023. The interval between the vertical red lines indicates the main phase of the storm, starting at SSC on April 23 at 8:56 UT. At the time of the SSC, the IMF $B_z$ is at 0 nT which turned southward, and then later reached to the value of -20 nT between 18:00 UT to 21:00 UT. During this time, the $V_{sw}$ increased from 400 km/s and reached to the maximum value of

751 km/s at 21:39 UT on April 23. After 3 hours, the IMF $B_z$ turned northward and fluctuated between 0 nT and 20 nT until 01:20 UT before a second strong southward excursion on April 24. This second long-term southward turn in IMF $B_z$ is seen on April 24 at 1:21 UT. During this turn, a sharp decrease in the IMF $B_z$ value from 20 nT to around -34 nT is observed as shown in Figure 2a. The IMF $B_z$ remains southward for approximately 3 hours until the end of the main phase at 4:03 UT on April 24. During this period, the $V_{sw}$ kept at approximately 600 km/s until the main phase ended at 4:03. Figure 2b shows

a two-step geomagnetic storm, with SYM-H minima values of -179 nT (at 21:59 UT) and -233 nT (at 04:03 UT) on April 23 and 24, respectively. During the main phase, the IEF $E_y$ oscillates rapidly between -21 and 21 mV/m. During the solar wind-magnetosphere interaction process, the Kp index increased to 8.3, indicating an intense geomagnetic storm conditions. Following the strong geomagnetic storm, the Kp index dropped from 8.3 to 6.7, then surged to 8, before returning to a calm




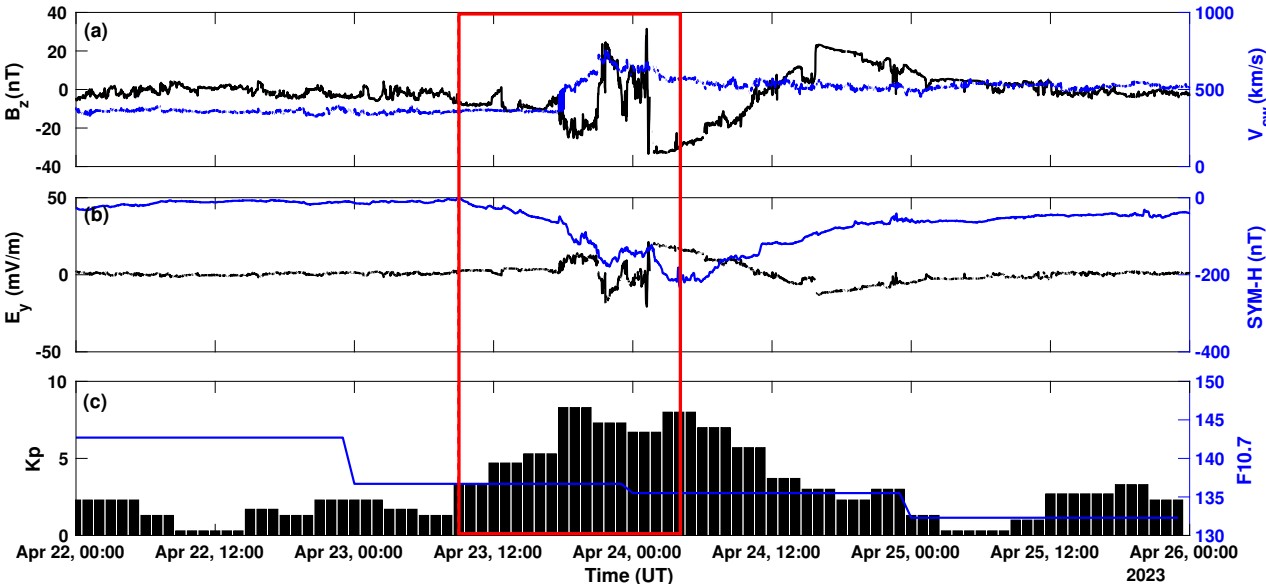

**Figure 2.** Temporal variation of: (a) solar wind speed ($V_{sw}$) and $B_z$ component of IMF, (b) $E_y$ component of IMF and $SYM-H$, and (c) the magnetic index ($K_P$) and the solar radio flux at 10.7 cm F10.7 during April 22-25, 2023 storm.

state. During this time, the solar activity level F10.7 remained steady at approximately 135 sfu as shown in Figure 2c. The

recovery phase began with the northward turning of the IMF $B_z$ and returned to the 0 nT after 24 h. During recovery phase, the $V_{sw}$ also gradually decreased to normal conditions.

**November 4-7, 2023 storm:** The main phase of this storm began with a southward turning of the IMF $B_z$, reaching to a lowest value of approximately -24.9 nT at 11:31 UT on November 5. Then at 12:00 UT it turned northward and reached maximum value of approximately 40.1 nT at 13:56 UT on the same day. Afterwards, the IMF $B_z$ shows a second long-term

southward turn reaching -20 nT until the end of the main phase at 16:54 UT. The $B_z$ component increased gradually to normal values on November 5. During the main phase, the $V_{sw}$ is at moderate level around 500 km/s and it increased to 600 km/s at 12:00 UT on November 6. Figure 3 (b) shows the IEF $E_y$ and SYM-H, correspondingly. During the compression phase, the SYM-H sharply increased to 23 nT at 17:09 UT on November 4 and shortly after this the main phase of the storm begins with decrease in SYM-H. It was also a double step storm with minimum values of SYM-H of approximately -100 nT (at 12:00 UT)

and -189 nT (at 16:54 UT) observed on November 5. Following the polarity of the IMF $B_z$, the IEF Ey also show fluctuations between -19.5 and 10.5 mV/m. The solar radio flux remained in the range $140 < F10.7 < 160$ sfu during this period. Before the SSC, Kp was less than 3+, indicating the quiet conditions, but as the storm begins, it rapidly increased to 5+ due to the initial effect of solar wind shock at 17:40 UT on November 4. From 00:00 to 08:00 UT on November 5, the Kp decreased to





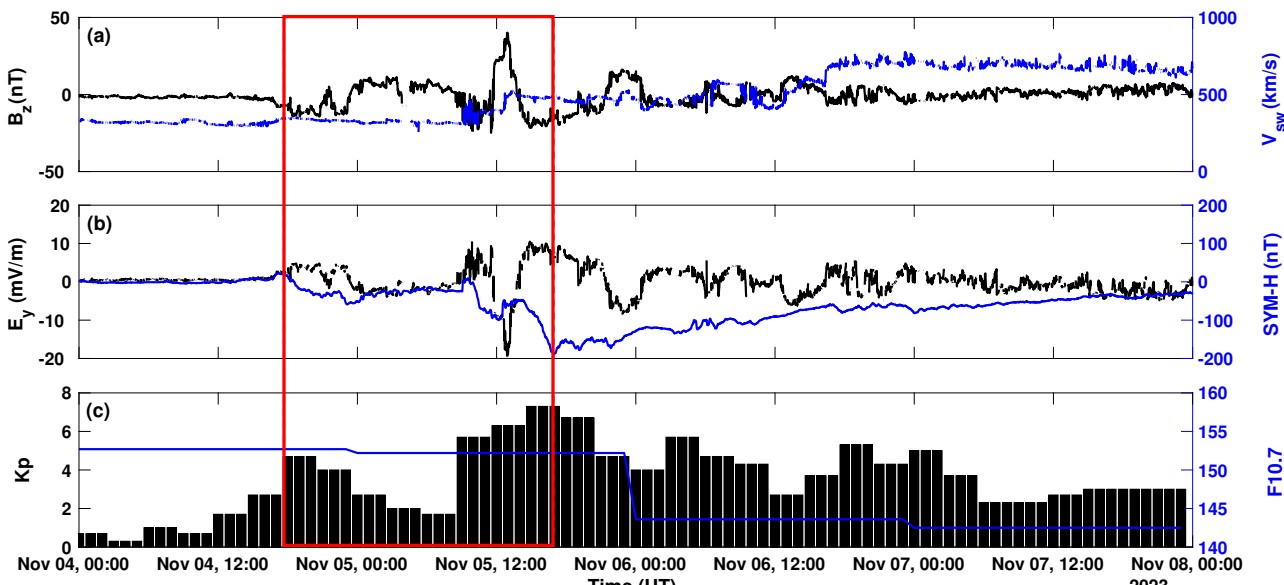

**Figure 3.** Temporal variation of parameters shown as: (a) solar wind speed ($V_{sw}$) and $B_z$ component of IMF, (b) $E_y$ component of IMF and $SYM - H$, and (c) the magnetic index ($K_P$) and the solar radio flux at 10.7 cm (F10.7) during November 4-7, 2023.

normal levels before rapidly increasing to 7+, predicting an intense geomagnetic storm between 12:00 UT and 19:00 UT on
November 5. The recovery phase started at 16:54 UT with a northward turning of the IMF $B_z$ on November 5.

**May 10-13, 2024 storm:** Figure 4a displays the temporal changes in $V_{sw}$ and the IMF $B_z$ during May 10 to 13, 2024. The
main phase began around 17:15 UT on May 10, with a sharp southward swing in the IMF $B_z$, reaching -30 nT, and an increase
in $V_{sw}$, reaching around 700 km/s. Figure 4b shows variations in SYM-H and IEF $E_y$ between May 10 and 13. Initially, the
SYM-H index increases sharply to 88 nT as the Earth's magnetosphere compresses at 17:15 UT on May 10. Following this
initial increase, the main phase begins with a reduction in the SYM-H due to increased ring current in the magnetosphere. The
main phase lasted until 02:14 UT on May 11, when the SYM-H index reached a maximum low of roughly -518 nT. During
this phase, the IEF $E_y$ fluctuates rapidly between -37 and 35 mV/m. The Kp index indicated substantial geomagnetic activity
on May 10 between 12 and 15 UT, reaching a value of 5. Shortly after the SSC, Kp reached to 8 and then began fluctuating
between 8 and 9 until the main phase ended at 02:14 UT on May 11. The solar activity level F10.7 stayed around 220 and 226
sfu as shown in Figure 4c. Following the storm recovery, the IMF $B_z$ shows rapid north-south oscillations while the SYM-
H increases gradually to reach the normal level. During this phase, the $V_{sw}$ increased to 998 km/s and the Kp index varied
between 7 and 9 until late evening on May 11.





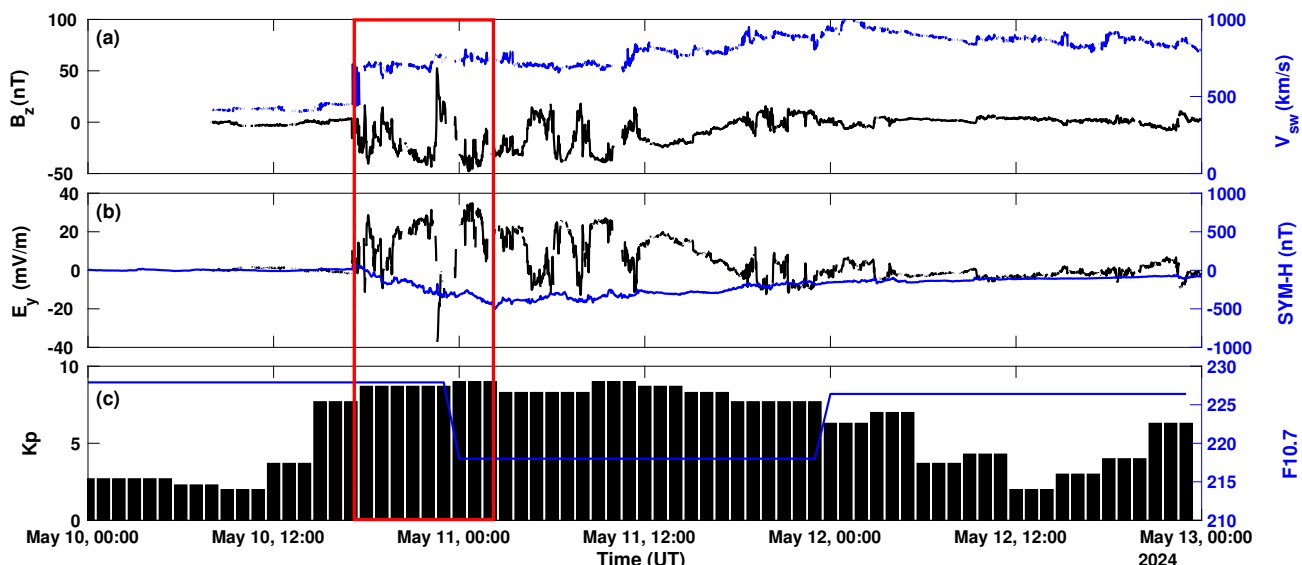

**Figure 4.** Temporal variation of: (a) solar wind speed ($V_{sw}$) and $B_z$ component of IMF, (b) $E_y$ component of IMF and $SYM - H$, and (c) the magnetic index ($K_p$) and the solar radio flux at 10.7 cm (F10.7) during May 10-13, 2024.

## 3.2 High-Latitude Ionosphere

During geomagnetic storms, an increased convective electric field and the massive injection of high-energy particles result in an
increase in Pederson currents and ionospheric conductivity (Buonsanto, 1999; Lukianova, 2020; Dungey, 1961; Axford, 1964).
This leads to an increase in the energy dissipation or Joule heating (JH), particularly in the Auroral zone where precipitation
of high-energy particles occurs. Enhanced Joule heating at auroral latitudes expands the neutral atmosphere, driving equatorial
neutral wind and generating traveling atmospheric/ionospheric disturbances (Richmond and Matsushita, 1975; Fuller-Rowell
et al., 1994). In order to estimate the solar wind energy dissipated into the inner magnetosphere-ionosphere (MI) system
through JH during the four severe geomagnetic storms SC25, we used the CCMC-hosted Weimer 2005 model. The quiet
time JH value varies from 1 to 10 $mW/m^2$. However, the value of JH increases significantly during magnetically disturbed
periods. During the main phase of each storm, the values of JH after sunset between the time 18:00 to 00:00 UT are given
in Table 3 and also shown in Figure 5. For the geomagnetic storm of March 23 to 25, 2023, JH values showed notable inter-
hemispheric asymmetries, with higher values in the southern hemisphere during the main phase. This storm was marked by
significant solar wind and IMF $B_z$ fluctuations, with IMF $B_z$ turning southward to -20 nT and $V_{sw}$ peaking at 600 km/s during
the recovery phase. The geomagnetic storm from April 23 to 25, 2023, exhibited significant IMF $B_z$ and $V_{sw}$ fluctuations,





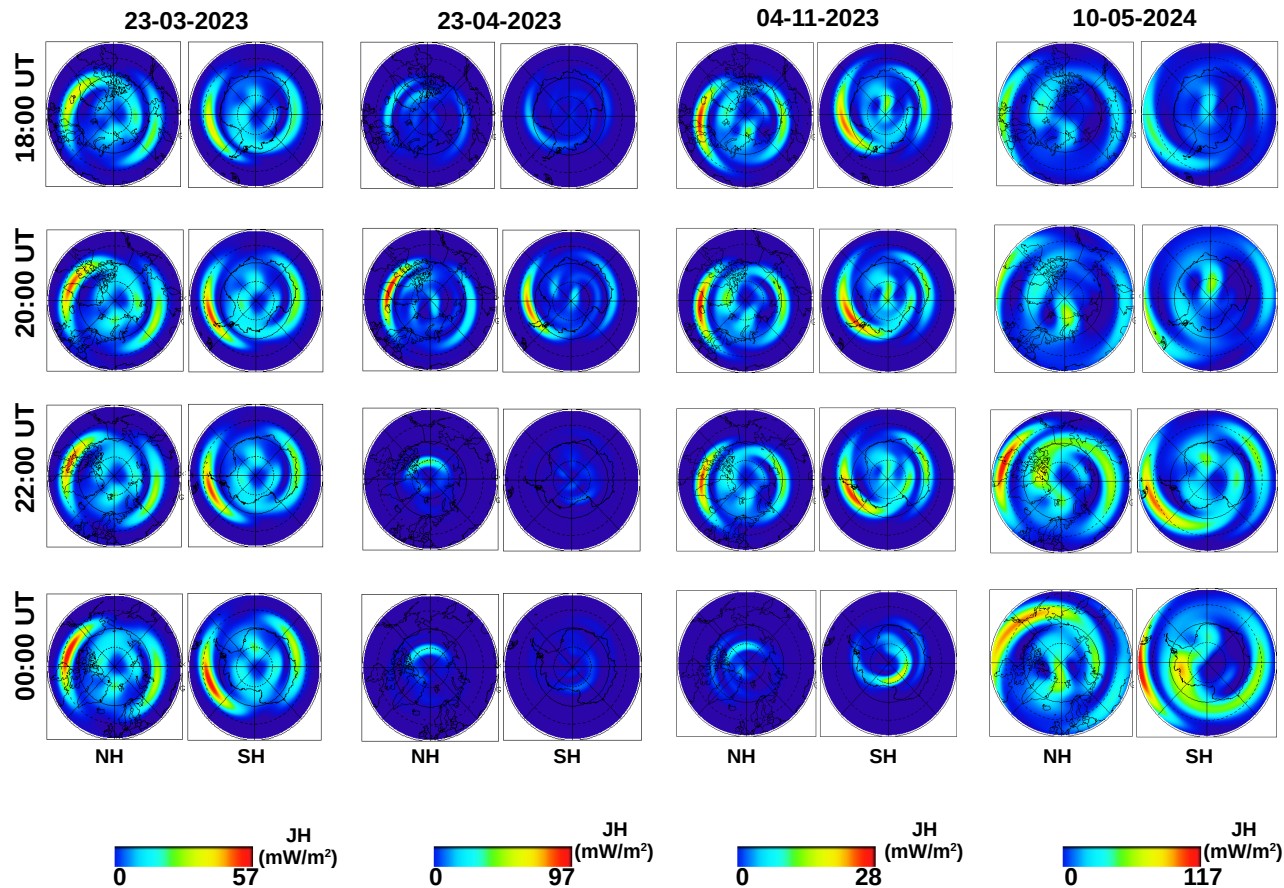

**Figure 5.** Polar maps of Joule heating from the Weimer 2005 Model computed at CCMC during the main phase of the four geomagnetic storms after sunset, from 18:00 UT to 00:00 UT.

leading to a two-step storm with SYM-H minima of -179 nT and -233 nT. JH values during this period showed significant inter-hemispheric asymmetries, with higher values in the northern hemisphere, particularly at 19:00 UT and 20:00 UT on April 23. The November 5-7, 2023 storm had notable IMF $B_z$ fluctuations, with the $V_{sw}$ increasing to 600 km/s during the main phase. JH values during this storm were relatively balanced between the hemispheres, with slight variations observed at different times. The May 10-13, 2024 storm was particularly intense, with the IMF $B_z$ reaching -30 nT and the $V_{sw}$ peaking at 700 km/s. JH values during this storm were significantly higher, especially in the northern hemisphere, with a peak of 114.7 mW/m$^2$ at 22:00 UT on May 10, indicating extreme geomagnetic activity and substantial energy dissipation in the ionosphere. In short, a strong JH is recorded during the main phase of the May 2024 storm from 18:00 to 00:00 UT. The order of JH after sunset from 18:00 to 00:00 UT is the following: May, March, April, and November. The inter-hemispheric asymmetries in the JH during distinct geomagnetic storms are caused by differences in heating between the northern and southern hemispheres. Factors such as Earth's magnetic dipole tilt, magnetic field geometry, seasons, and the orientation of the IMF, in particular the





IMF $B_y$ (east-west component), all contribute to the JH's inter-hemispheric asymmetry (Smith et al., 2023). The storm-time Joule heating in high-latitude can cause atmospheric gravity waves (AGWs) to move equatorward, thus providing favorable
conditions for formation equatorial/low-latitude ionospheric plasma irregularities (H Jin, 2022).

### 3.3 Low-Latitude Ionosphere Response

During magnetic storms, the low latitude ionosphere exhibits different features due to the complicated interplay of geomagnetic activity, solar radiation, and ionospheric dynamics. Some of these features are discussed here:

### 3.3.1 Structure of Equatorial Ionization Anomaly (EIA)

Figure 6 (a-f) show the temporal variation of the ionospheric TEC during the main phase of the March 23-24, 2023 geo-magnetic storm after sunset, from 18:00 UT to 00:00 UT. During the March 23-25, 2023 storm, TEC values showed notable inter-hemispheric asymmetries, with higher values in the southern hemisphere during the main phase. Initially, well-separated double crests in the EIA were observed, but merging of these crests occurred between 18:00 and 20:00 UT, leading to increased TEC in the equatorial zone. This merging and subsequent increase in TEC were driven by enhanced JH and geomagnetic activ-
ity. At the same time, the TEC began to gradually increase in the northern hemisphere. This increase continued, and by 20:00 UT, TEC values were higher in the northern hemisphere than in the southern hemisphere. It is also observed that in the southern hemisphere, the TEC anomalies are confined to low latitudes and they extend to mid-latitudes in the northern hemisphere. At 22:00 UT, the TEC decreases in the equatorial zone which again leads to the formation of the double crests in the EIA. On March 24, from 00:00 UT to 06:00 UT, high TEC values were observed at equatorial and low latitudes, indicating geomagnetic
disturbances towards the Asian sector. At 12:00 UT, the double crest in the northern hemisphere became a single crest, and TEC values slightly decreased. Starting from 14:00 UT, ionospheric TEC values began to increase again, marking the begin-ning of a second phase of ionospheric irregularities. The increase in TEC values towards the northern hemisphere indicated a period of geomagnetic disturbance until 22:00 UT. The decreasing TEC values are observed during the recovery phase of the geomagnetic storm from 06:00 UT onwards (See supporting Figure of madrigal TEC maps for March 23-24, 2023).
Figure 6 (g-i) depicts the temporal evolution of the ionospheric TEC during the strong geomagnetic storm of April 23-24, 2023. The April 23-24, 2023 storm exhibited well-defined double crests in the EIA on April 23, with TEC increasing in the EIA crests over America and expanding poleward due to strong JH. After achieving the peak value for the day, the TEC begins to decrease from 22:00 UT forward. The EIA's strength and latitudinal range had significantly decreased, as evidenced by the global ionospheric TEC distribution on April 24. The main phase of this geomagnetic storm terminates at 04:00 UT, with the
EIA's double crest over the southern hemisphere decreasing to a single crest (See the supporting Figure of madrigal TEC maps for April 23-24, 2023). Following the peak TEC value in the southern crest, a recovery phase begins at 4:03 UT and persisted until 22:00 UT. During the recovery phase, the ionospheric TEC returns to normal state.

Figure 6 (m-r) shows ionospheric TEC maps during show the temporal variation of the ionospheric TEC during the main phase of the November 4-5, 2023 geomagnetic storm after sunset, from 18:00 UT to 00:00 UT. The November 4-5, 2023 storm saw
storm-time ionization initially confined to the equatorial region, resulting in underdeveloped EIA crests. However, by 20:00



**Figure 6.** Madrigal-based global ionospheric TEC maps during the main phase of the four geomagnetic storms after sunset, from 18:00 UT to 00:00 UT shown as: (a-f) March 23-24, 2023, (g-l) April 23-24, 2023, (m-r) November 4-5, 2023, and (s-x) May 10-11, 2024.

UT, ionization increased in the northern and southern low latitudes, forming a weak EIA with smaller latitudinal extent.
On May 10, 2024, a double crest EIA structure at the equator containing normal TEC values with minor variations observed in other regions until 18:00 UT (See the supporting Figure of madrigal TEC maps for May 10-11, 2024). The global ionospheric maps in Figure 6 (s-x) illustrate the temporal variation of the ionospheric TEC during the main phase of the May 10-11, 2024 geomagnetic storm after sunset, from 18:00 UT to 00:00 UT. The May 10-11, 2024 storm was particularly intense, with substantial increases in TEC in the EIA crests over American longitudes and significant poleward shifting of the EIA crests up to about ±35° geomagnetic latitudes, associated with an external fountain effect. The recovery phase shows negative phase of this storm with a strong reduction in TEC along with EIA crests suppression and merging into a single equatorial band (See the supporting Figure of madrigal TEC maps for May 10-11, 2024). In each case, the variations in TEC were influenced



by the intensity and duration of geomagnetic activity, driven by solar wind parameters and enhanced JH, which affected the distribution and magnitude of ionospheric disturbances.

### 3.3.2   Ionospheric Plasma Irregularities

Figures 7-10 show temporal profiles of ionospheric plasma irregularities in the three longitudes that are America, Africa, and
Asia during the 4 geomagnetic storms. Each storm's unique solar wind conditions, geomagnetic activity, and JH distribution influenced the extent and location of ionospheric disturbances and plasma bubbles. During the March 23-25, 2023 storm, strong post-sunset plasma irregularities were observed in low/equatorial latitude stations (BOGT and RIOP) in the American sector, driven by significant solar wind and IMF $B_z$ fluctuations, and enhanced JH. In contrast, the April 23-25, 2023 storm also showed post-sunset plasma irregularities over the same stations, but no irregularities were noted in other regions, highlighting the
localized impact of JH and geomagnetic activity. The November 4-6, 2023 storm saw highly suppressed post-sunset ionospheric plasma irregularities across all sectors due to lower ionospheric conductivity and weak electric fields, which reduced vertical plasma drifts and prevented irregularities from growing. The severe May 10-13, 2024 storm had a significant impact on the ionosphere, causing pronounced post-sunset plasma irregularities, particularly over southern low latitude stations (CORD and SANT) in the American sector. This was due to JH's inter-hemispheric asymmetry, where greater JH in the northern hemisphere
led to the penetration of atmospheric gravity waves (AGWs) into the southern hemisphere, resulting in higher post-sunset plasma irregularities at southern low latitudes.

The distribution and density of plasma are affected by the changes in ionospheric electric fields and currents resulting from DD electric field caused by storm-time neutral winds and PPEF of magnetospheric origin (Astafyeva et al., 2018). Figure 11 (a-d) depicts the temporal fluctuations of the PPEF during geomagnetic storms on March 22-25, 2023, April 22-25, 2023,
November 4-7, 2023, and May 10-12, 2024. At 18:00 UT during the main phase of the March 23 storm, a westward PPEF of magnitude -0.55 mV/m was seen in the Asian sector. At the same time, an eastward PPEF of around 0.5 mV/m can be observed in the American and African sectors before dusk. Following sunset on April 23, multiple PPEFs of variable duration are visible in the three sectors. In America, the four PPEFs are noticeable. The first peak value is 0.33 mV/m from 18:05 to 20:05 UT, the second is -0.78 mV/m from 18:05 to 20:05 UT, the third is 0.61 mV/m from 23:05 to 23:55 UT, and the fourth
is 0.45 mV/m from 00:40 to 03:15 UT. In the African sector, two PPEFs of varied duration with maximum values of roughly 0.60 mV/m and -0.65 mV/m are observed before midnight from 21:00 UT to 23:00 UT and 23:05 to 23:55 UT. A strong long-duration westward PPEF with a peak value of around -1.36 mV/m is also detected after midnight, between 1:45 and 4:35 UT. In Asia, the first westward PPEF with a peak value of -0.54 mV/m occurs from 18:05 to 20:05 UT, the second eastward is 0.94 mV/m from 21:10 to 23:00 UT, and the third long duration eastward PPEF with a peak value of 0.66 mV/m occurs from
01:45 to 04:35 UT on April 24. During the main phase of November 4-7, 2023 storm, the PPEFs of different polarities and duration reach the equatorial ionosphere during the day. In America, the first westward PPEF with maximum value of about -0.32 mV/m occurred in the morning hours from 8:30 to 10:25 UT, and the second eastward penetrating field of magnitude about 0.44 mV/m is observed around the noon from 12:05 UT to 13:25 UT. In Africa, two noticeable PPEFs with peak values



of -0.53 mV/m and 0.34 mV/m are observed around noon. In Asia, there is one noticeable westward penetrating field with a
peak value of -0.42 mV/m from 12:05 to 13:35 UT. During mother's day geomagnetic storm on May 10–12, 2024, we observe
significant storm-time PPEFs after sunset. For instance, intense eastward-directed PPEFs of different durations between 18:00
and 22:00 UT causes super fountain effect in the American sector. This results in to a notable variations in ionospheric electron
density and the positioning of EIA crests as shown in madrigal TEC maps. We can also see eastward penetrating electric fields
with maximum values of 2.59 mV/m and 1.80 mV/m in Africa and Asia, respectively, and westward PPEF with a peak value
of roughly -2.73 mV/m in America between 22:50 and 23:25 UT. America and Africa experience an eastward PPEF with a
peak value of 1.39 mV/m and a westward PPEF with a peak value of -1.95 mV/m, respectively, between 23:30 and 2:00 UT.
During the March 23-25, 2023 storm, a westward PPEF of -0.55 mV/m was observed in the Asian sector at 18:00 UT, while
an eastward PPEF of 0.5 mV/m was seen in the American and African sectors before dusk. These PPEFs contributed to the
merging of EIA crests and increased TEC in the equatorial zone, leading to strong post-sunset plasma irregularities in the
American sector. The April 23-25, 2023 storm exhibited multiple PPEFs of varying magnitudes and durations across different
sectors, with significant eastward and westward PPEFs in America, Africa, and Asia. These PPEFs influenced the expansion of
EIA crests and increased TEC, particularly in the northern hemisphere, and caused noticeable post-sunset plasma irregularities
in the American sector. The November 4-6, 2023 storm had PPEFs of different polarities and durations, with westward and
eastward PPEFs observed in America, Africa, and Asia. However, lower ionospheric conductivity and weak electric fields
suppressed post-sunset plasma irregularities across all sectors. The May 10-13, 2024 storm featured intense eastward-directed
PPEFs between 18:00 and 22:00 UT, causing the super fountain effect in the American sector, leading to significant TEC
variations and poleward shifting of EIA crests. Strong eastward and westward PPEFs were also observed in Africa and Asia,
contributing to enhanced post-sunset plasma irregularities, particularly in the southern low latitude stations.

## 4    Conclusions

The comprehensive analysis of the geomagnetic storms from March 23-25, April 23-25, November 4-6, 2023, and May 10-13,
2024, reveals that solar wind parameters, geomagnetic activity, JH, and PPEFs significantly influence ionospheric TEC varia-
tions, EIA crest formations, and post-sunset plasma irregularities. Each storm exhibited unique patterns of TEC changes, driven
by the intensity and duration of geomagnetic disturbances and JH distribution. The storms also highlighted the importance of
inter-hemispheric asymmetries in JH, which affected the distribution and magnitude of ionospheric disturbances. Recom-
mendations include enhancing monitoring and predictive models for solar wind and geomagnetic activity to better anticipate
ionospheric responses, improving communication and navigation systems' resilience to geomagnetic storms, and conducting
further research on the mechanisms driving inter-hemispheric asymmetries and their impact on ionospheric dynamics.

*Data availability.* Solar wind parameters and magnetic indices are obtained from the GSFC/SPDF OMNIWeb interface (https://omniweb.
gsfc.nasa.gov/form/omni_min.html). MIT-TEC maps are obtained from the MIT Haystack website (http://cedar.openmadrigal.org/static/





**Table 3.** Joule Heating (JH), and the Equatorial Ionization Anomaly (EIA) during the main phase of the four geomagnetic storms after sunset, from 18:00 UT to 00:00 UT.

| Storm Days | Time (UT) | JH($mW/m^2$) Northern Hemisphere | JH($mW/m^2$) Southern Hemisphere | EIA structure and Ionization level |
|---|---|---|---|---|
| March 23, 2023 | 18:00 UT | 41.3 | 41.8 | merged or single crest (moderate) |
| – | 19:00 UT | 40.7 | 40.7 | merged or single crest (strong) |
| – | 20:00 UT | 50.9 | 52.6 | merged or single crest (strong) |
| – | 21:00 UT | 52.5 | 53.4 | merged or single crest (strong) |
| – | 22:00 UT | 50.1 | 52.0 | merged or single crest (strong) |
| – | 23:00 UT | 52.2 | 53.9 | double crest (moderate) |
| March 24, 2023 | 00:00 UT | 56.6 | 55.9 | double crest (moderate) |
| April 23, 2023 | 18:00 UT | 33.7 | 22.9 | double crest (moderate) |
| – | 19:00 UT | 94.7 | 77.4 | double crest (strong) |
| – | 20:00 UT | 97.0 | 85.4 | double crest (strong) |
| – | 21:00 UT | 73.5 | 52.2 | double crest (strong) |
| – | 22:00 UT | 44.7 | 14.4 | double crest (strong) |
| – | 23:00 UT | 55.1 | 24.6 | double crest (moderate) |
| April 24, 2023 | 00:00 UT | 37.0 | 13.2 | double crest (weak) |
| November 4, 2023 | 18:00 UT | 26.4 | 24.3 | merged or single crest(weak) |
| – | 19:00 UT | 30.1 | 27.7 | merged or single crest (weak) |
| – | 20:00 UT | 27.9 | 25.6 | merged or single crest (weak) |
| – | 21:00 UT | 22.4 | 22.0 | merged or single crest (weak) |
| – | 22:00 UT | 25.2 | 26.0 | double crest (weak) |
| – | 23:00 UT | 25.0 | 28.0 | double crest (weak) |
| November 5, 2023 | 00:00 UT | 10.3 | 19.7 | double crest (weak) |
| May 10, 2024 | 18:00 UT | 63.8 | 51.8 | double crest (moderate) |
| – | 19:00 UT | 47.7 | 30.4 | double crest (moderate) |
| – | 20:00 UT | 83.1 | 80.9 | double crest (strong) |
| – | 21:00 UT | 74.5 | 80.2 | double crest (strong) |
| – | 22:00 UT | 114.7 | 101.4 | double crest (strong) |
| – | 23:00 UT | 57.1 | 73.2 | double crest (strong) |
| May 11, 2024 | 00:00 UT | 98.5 | 116.2 | double crest (weak) |



experiments4/2021/gps/). Global ROTI maps are available at https://stdb2.isee.nagoya-u.ac.jp/GPS/GPS-TEC/. ROTI variations videos are obtained from System for Ionosphere Monitoring and Research from GNSS database, at (https://simurg.iszf.irk.ru/create_map). Equatorial electric fields over a specific longitude are obtained from the real-time Prompt Penetration Equatorial Electric Field Model (PPEEFM) available at https://geomag.colorado.edu/real-time-model-of-the-ionospheric-electric-fields.html. Weimer model Simulations are provided by the Community Coordinated Modeling Center (CCMC) at NASA Goddard Space Flight Center through (https://ccmc.gsfc.nasa.gov/

models/).

*Competing interests.* It is declare that the authors have no competing interest.





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




**Figure 7.** Temporal variation of ionospheric ROTI during March 22-25, 2023.







**Figure 8.** Temporal variation of ionospheric ROTI during April 22-25, 2023.





**Figure 9.** Temporal variation of ionospheric ROTI during November 4-7, 2023.





**Figure 10.** Temporal variation of ionospheric ROTI during May 10-13, 2024.





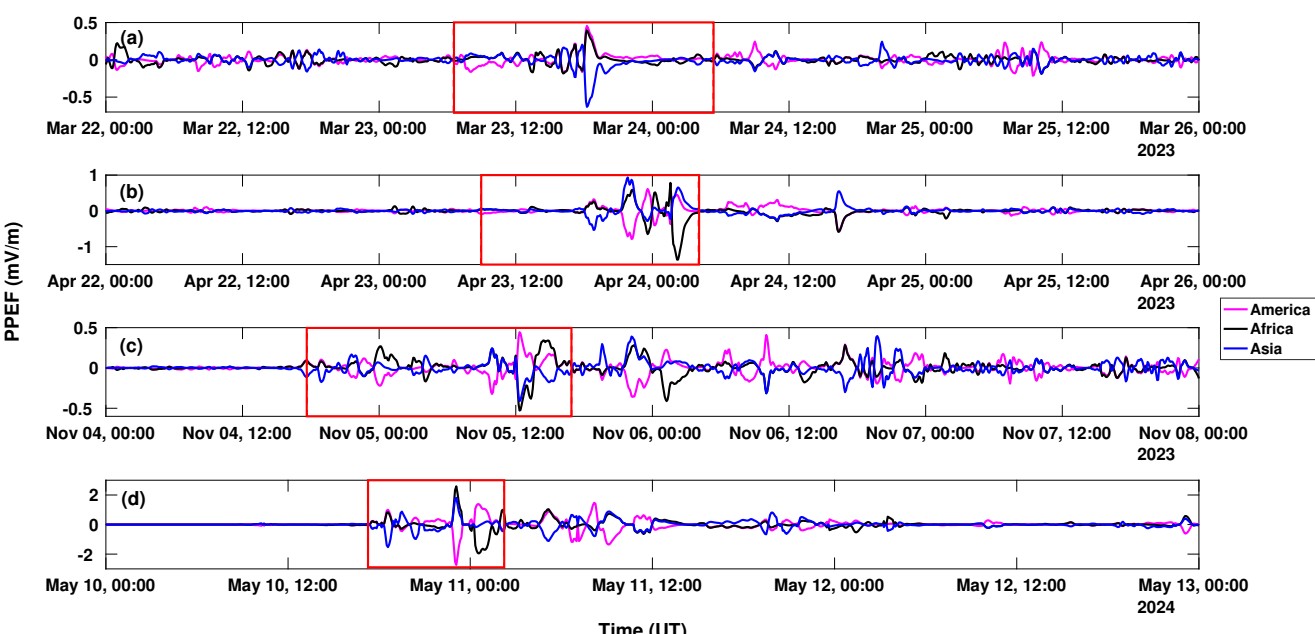

**Figure 11.** Variations in the ionospheric prompt penetration electric fields (PPEFs) over the three longitudes during the three intense geo-magnetic storms of SC 25.