# Peer review of "Ionospheric Plasma Irregularities During Intense geomagnetic storms of Solar Cycle 25"

_EGUsphere, 2025_

## Referee Comment (RC2)

Comments on the manuscript entitled '**Ionospheric Plasma Irregularities During Intense geomagnetic storms of Solar Cycle 25**' authored by Nadia Imtiaz et al. (2025) to ANGEO (manuscript egusphere-2025-86).

**General comments:**

This study presents the characteristics of equatorial and low latitude ionosphere during four intense geomagnetic storms that occurred on March 23–25, 2023, April 23–25, 2023, November 4–7, 2023 and May 10–13, 2024 in the ascending phase of Solar Cycle 25 based Joule Heating (JH) rates at high latitudes. For this purpose, authors have utilized the Weimer 2005 Model simulations to analyse the JH patterns along with Madrigal TEC maps to identify the changes in the intensity, location, and symmetry of the equatorial ionization anomaly (EIA) during these intense geomagnetic storms. Equatorial/low-latitude ionospheric plasma irregularities at different longitudes under geomagnetically disturbed conditions are studied using the TEC derived ROTI measurements. While a strong JH is observed during the May 2024 storm in the main phase, the other storms have the following order from strong to weak: March 2023, April 2023, and November 2023. The authors have investigated especially the role of asymmetric Joule heating in the structuring of the EIA like double crest, single crest or merged. Authors have utilized these features to investigate the formation or suppression of equatorial plasma irregularities. It is suggested that the generation of these ionospheric plasma irregularities and their latitudinal distribution strongly dependent on EIA's density gradients and latitudinal density structure. For instance, while the double crest EIA structures with strong plasma density gradients playing an important role in the generation of post sunset ionospheric plasma irregularities during the main phases of these four geomagnetic storms, single crest or merged EIA structure didn't favour the generation of ionospheric plasma irregularities. In addition, the role of storm-time penetration electric field in the structuring and seeding of ionospheric plasma irregularities have been investigated. Analysis of these storms by authors suggest that solar wind parameters, geomagnetic activity, JH, and PPEFs significantly influence ionospheric TEC variations, EIA crest formations, and post-sunset plasma irregularities. These investigations revealed that these storms highlighted the importance of inter-hemispheric asymmetries in JH that affected the distribution and magnitude of ionospheric irregularities. Finally, the paper concludes that the resulting change in the thermospheric winds and electric fields due to storm conditions can alter the EIA structures which will impact the plasma irregularities at equatorial and low latitudes.

While many of the results presented here are already reported, I believe discussions like role of single EIA, double or Merged EIA on plasma irregularities looks to be interesting though it is known that asymmetry of EIA can suppress the plasma bubbles due to thermospheric meridional winds. I am providing the comments to the authors. The manuscript may be considered for publication only after implementing the comments made in this review report by incorporating appropriate changes.

**Specific comments**

- It is mentioned that JH is strong in May 2024 Mother's Day storm during main phase. Then it followed like this: March 2023>April 2023>November 2023. Cause of variation proposed: Change in thermosphere winds and electric fields due to storm conditions alter EIA structures. What is the role of thermospheric neutral density such as O/N2 ratio on the EIA single crest, double crests?
- Whether authors have investigated positive/negative storms on the plasma irregularities
- How the symmetry/asymmetry of EIA with single, double crests can influence the EPBs are not discussed. Please provide discussions.
- What is the role of electric fields on the generation/suppression of plasma irregularities?
- Why authors have chosen TEC maps between 18:00-00:00 UT for all the storm events in Figure-6? Since different storms have main phases at different UT times, why authors have chosen this time interval to study the plasma irregularities and their connection to EIA.

- The plots shown in Figure-6 can be plotted as a line plots to show clear asymmetry in EIA. Also in the plot, map for 23:00 UT is not shown. Whether it is plotted like this only, please clarify.
- Role of geomagnetic storms on plasma irregularities in the following papers: S. Sripathi and Ram Singh, A study on the response of the ionosphere to the three major space weather events of 2015 using meridional chain of ionosondes and GPS receivers over India, journal of Sun and Geosphere, DOI: 10.31401/SunGeo.2018.02.08, 2020
- Similarly, the following paper discusses the role of geomagnetic storms and role of Es layers on the plasma irregularities. Please go through it. Singh, R., & Sripathi, S. (2020). A statistical study on the local time dependence of equatorial spread *F* (ESF) irregularities and their relation to low-latitude Es layers under geomagnetic storms. *Journal of Geophysical Research: Space Physics*, *125*, e2019JA027212. https://doi.org/10.1029/2019JA027212
- It is said that the generation of ionospheric plasma irregularities and geographical distribution strongly dependent on EIA density gradients and general density structure. Please include the physics of their dependency.
- It is mentioned that the storm-time Joule heating in high-latitude can cause atmospheric gravity waves (AGWs) to move equatorward, thus providing favourable conditions for formation equatorial/low-latitude ionospheric plasma irregularities. But it is under recovery phase mostly. But here authors are describing plasma irregularities in the main phase. Please clarify.
- Double crest EIA structures with strong plasma density gradient play an important role in generation of post sunset plasma irregularities. Single crest or merged EIA does not favour the generation of irregularity. Please provide brief explanation for this. How the plasma irregularity generation is associated with EIA crest need to be mentioned clearly in the manuscript.
- 3.3.1 Structure of Equatorial Ionization Anomaly (EIA): "This merging and subsequent increase in TEC were driven by enhanced JH and geomagnetic activity". The claim can be elaborated by explaining the variation of JH with EIA structure.
- In general, PPEFs will affect the equatorial and low latitudes in the main phase of the storm, while Joule heating will affect the equatorial and low latitudes in the recovery phase of the storm through thermospheric wind circulation. However, here, authors are discussing the role of Joule heating during Main phase which I am not able to understand. Please clarify how the Joule Heating can impact the EIA crest in the main phase.
- The EIA trends are visually explained in table-3 with JH values where pattern can be observed. However, there is no mention of physics of their connection with the JH. What is the relation between JH and EIA dynamics are not clearly mentioned. Please explain. Similarly, Table-3 doesn't show any location where EIA crest is strong/weak. It has to be clearly mentioned.
- 3.3.2 Ionospheric Plasma Irregularities: The observations presented here under PPEFs and DDEFs and claims to have plasma irregularities. The proof/data/evidence/observation of the irregularities are needed to be included.
- What is the role of atmospheric waves and tides on the variability of the PRE, EIA and plasma irregularities. It is known that equatorial electrodynamics plays significant role in day-to-day variability of the E X B drift and RT instability triggered plasma irregularities. So, this point needs to be addressed.
- Authors have chosen these four storms without looking into the role of seasons on the EIA. Usually, it is standard practice to choose storms that fall in equinox. Please address this point.
- If you want to study only 18:00-00:00 UT, please show this period with yellow highlight in the Figures-1-4 so that it is easy for the readers to follow it. Currently, authors have highlighted the main phase of the storm with 'red' rectangle box. In fact, in these figures, AE/AU/AL is missing.
- In the introduction, the sentence starts with 'Ionospheric effects on applications are generally minor in mid-latitudes...' may be replaced with 'Ionospheric effects are generally minor in mid-latitudes...' as the sentence is not conveying message correctly.

- "It is generally said that during storm time, it increases the daytime eastward electric field". $E_{total}(t)=E_{quiet}(t)+E_{PPEF}(t)$. It can either increase or decrease the equatorial zonal electric field, depending on IMF Bz polarity and whether you are on the dayside or nightside. This equation can be included.
- Correct 'Disturbed Dynamo (DD)' to 'Disturbance Dynamo (DD)'

---

## Author Comment (AC1)

Dear Editor,

Thank you for sending us the report on "Ionospheric Plasma Irregularities During Intense geomagnetic storms of Solar Cycle 25" manuscript No. egusphere-2025-86 submitted to Annales Geophysicae. We are very grateful to the Editor and anonymous referee for the time and efforts invested in reviewing our work. We have incorporated the general comment provided by one reviewer in this round.

**Response to Reviewer:**

**Major comments:**

**The authors present ROTI plots in Universal Time (UT), which makes it difficult to interpret the local time dependence of irregularities. Since ionospheric phenomena like post-sunset irregularities are strongly influenced by local time, this choice limits the interpretability of the results. Local time (LT) plots would better visualize the storm-time impact and allow comparisons across different longitude sectors. Therefore, the authors should replot their ROTI in Local Time (LT), even though they provided LT information in table 1, to better capture the local time dependence of irregularities. This would allow them to identify whether irregularities are enhanced or suppressed during specific local times (e.g., post-sunset, post-midnight) and also correlate storm impacts with the local time of SSC and main phase onset. The authors also mention plasma irregularity suppression during certain geomagnetic storms (e.g., November 2023), this would be good in terms of local time discussions. Several studies identified the importance of the local time at which geomagnetic storms happen to explain their impact on the ionosphere. Please review these documents to gain a better understanding of the**

above, including discussions related to DDEF, PPEF, and the suppression of irregularities.

1. Liu, J., Zhao, B., & Liu, L. (2010, March). Time delay and duration of ionospheric total electron content responses to geomagnetic disturbances. In Annales Geophysicae (Vol. 28, No. 3, pp. 795-805). Copernicus GmbH.

2. Amaechi, P. O., Oyeyemi, E. O., & Akala, A. O. (2018). Geomagnetic storm effects on the occurrences of ionospheric irregularities over the African equatorial/low-latitude region. Advances in Space Research, 61(8), 2074-2090.

3. Araujo-Pradere, E. A., Fuller-Rowell, T. J., Codrescu, M. V., & Bilitza, D. (2005). Characteristics of the ionospheric variability as a function of season, latitude, local time, and geomagnetic activity. Radio Science, 40(05), 1-15.

4. Seba, E. B., & Nigussie, M. (2016). Investigating the effect of geomagnetic storm and equatorial electrojet on equatorial ionospheric irregularity over East African sector. Advances in space Research, 58(9), 1708-1719.

Reply: Following to the Referee's report, we have plotted ROTI in Local Time (LT) to identify whether irregularities are enhanced or suppressed during specific local times (e.g., post-sunset, post-midnight) and also correlated storm impacts with the local time of SSC and main phase onset. Please see the revised version of Figures 8-10 and Table 3. For better understanding, the discussion on the basis of above literature has been added in the revised manuscript. Following discussion regarding local time dependence of the post-sunset ionospheric plasma irregularities during the storms have been added in the Results and Discussion section.

The storm-time ring current is critical for both the development and suppression of equatorial ionospheric plasma irregularities. For instance, ionospheric plasma irregularities are generated if the greatest excursion of the ring current (minimum SYM-H index) occurs between midnight and postmidnight. However, the plasma irregularities are suppressed when the maximum excursion of the ring current occurs in the early afternoon (Aarons, 1991). Table 3 shows the LT of maximum excursion of ring current or MPE at each GNSS location. During geomagnetic storms in March, April, and May, the ring current has the greatest excursion after midnight in American sector, indicating the highest likelihood of plasma irregularities in the evenings following the storms' main phases in this sector. For instance, the maximum ring current excursions at the GNSS stations BOGT, RIOP, and SANT (LT = UTC-5) occurred at approximately 00:21, 23:03, 11:54, and 21:14 LTs on March 24, April 24, November 5, and May 11, respectively. On the same days, the CORD station (LT= UTC-4) reported the highest excursions at 01:21, 00:03, 12:54, and 22:14 LT. Notably, the highest excursions during the storms of March, April, and May occurred predominantly during the night to midnight which is the most ideal time for the growth of post-sunset ionospheric plasma irregularities. During November storm, the maximum ring current excursion occurred at approximately 11:54, and 12:54 LTs which is the least favorable time for ionospheric plasma irregularities to occur. However, the intensity of post-sunset plasma irregularities varies between these storms due to complex interaction of important factors such as storm-time electric fields, meridional winds, background ionospheric conditions, local time and seasons. The largest excursion of the ring current in the November storm

occurred before dusk, indicating that post-sunset plasma irregularities were unlikely during the main phase of this storm.

The authors attempt to link high-latitude Joule Heating (JH) with the Equatorial Ionization Anomaly (EIA), which is a low-latitude phenomenon (lines 240-250). While this is an interesting approach, the connection between these two regions is not clearly explained in the paper. The paper shows correlations between JH and EIA structures, but it does not establish a causal relationship. The authors should provide more evidence or references to support their claim that JH directly influences EIA and explicitly state the mechanism by which high-latitude JH influences low-latitude EIA. Do the authors propose that JH-driven thermospheric winds transport plasma or alter neutral composition, thereby affecting EIA structures? The authors observe interhemispheric asymmetries in JH (e.g., stronger JH in the northern hemisphere during the May 2024 storm). How do these asymmetries translate to differences in EIA structures or irregularities at low latitudes?

Reply: Joule heating has a considerable and primarily indirect effect on the post-sunset EIA during the main phase of a geomagnetic storms, which is mediated by storm-time neutral winds, thermospheric upwelling, and electric field modulation. Following to the Referee's suggestions, we have improved the Results and Discussion section.

The authors' discussion on the relationship between EIA structures and ionospheric irregularities is a good starting point, but it lacks several critical elements that would make their findings more robust, insightful, and convincing. That is, the authors do not provide a detailed mechanistic explanation of how specific EIA structures (e.g., double crest, single crest, or merged) influence the formation or suppression of irregularities. For example: How do density gradients in the EIA directly affect the night time

**growth of Rayleigh-Taylor instabilities? Please review the followings for further understanding:**

**• Luan, X. (2021). Equatorial ionization anomaly variations during geomagnetic storms. Ionosphere dynamics and applications, 301-312.**

**• Balan, N., Liu, L., & Le, H. (2018). A brief review of equatorial ionization anomaly and ionospheric irregularities. Earth and Planetary Physics, 2(4), 257-275.**

**• Seba, E. B., Nigussie, M., & Moldwin, M. B. (2018). The relationship between equatorial ionization anomaly and nighttime equatorial spread F in East Africa. Advances in Space Research, 62(7), 1737-1752.**

**• Aa, E., Chen, Y., & Luo, B. (2024). Dynamic expansion and merging of the equatorial ionization anomaly during the 10–11 May 2024 super geomagnetic storm. Remote Sensing, 16(22), 4290.**

**Reply:** Following to the Referee's suggestion, we have added detailed discussion about EIA structures, density gradients in EIAs and their connection with ionospheric plasma irregularities. Please see Section 3.3.2 of the revised manuscript.

**Suggestion on the conclusion section:**

**Here are my suggested potential findings that should be included in the conclusion section and summarized in the abstract, along with clear physical explanations (the physical explanations in discussion and conclusion).**

**Reply:** Following to the Referee's report, we have improved the conclusion section as:

- **The distribution and intensity of Joule heating during geomagnetic storms varied significantly with the storm's magnitude, local time of occurrence, duration, and seasons. Among the four storms examined, the May storm (near the June solstice) exhibited the most intense post-sunset Joule heating, followed by the March (equinox),**

April (post-equinox), and November (fall transition) storms, with the latter showing the weakest activity. Nearly symmetric heating patterns were observed during equinoctial storms, whereas solstice storms led to marked asymmetries between hemispheres.

- Each storm produced distinct equatorial ionization anomaly (EIA) structures, ranging from single crests with filled troughs to uneven double crests with deep troughs. These configurations were generated by the interaction between storm-induced electric fields and equatorward neutral winds, both of which are modulated by the spatial and temporal characteristics of Joule heating. After sunset, equatorward meridional winds (southward in the Northern Hemisphere and northward in the Southern Hemisphere) can lift low-latitude plasma upward along magnetic field lines, resulting in pronounced double-crest EIAs. In contrast, hemispherically asymmetric winds may amplify the EIA in one hemisphere while suppressing it in the other, forming a single crest. When poleward winds dominate in both hemispheres, plasma uplift is inhibited, leading to weaker EIA development.

- The combined influence of storm-time electric fields and thermospheric winds plays a key role in shaping the EIA, elevating plasma to higher altitudes, and redistributing electron density across trough and crest regions. During the main phase of storms, frequent North/South flipping in the interplanetary magnetic field (IMF Bz) can generate both eastward and westward prompt penetration electric fields (PPEFs), resulting from undershielding and overshielding effects. These alternating electric fields drive upward and downward vertical plasma drifts, further altering EIA structure.

- In particular, EIAs with filled or deep troughs and steep vertical gradients in plasma density can create favorable conditions for the development of Rayleigh-Taylor instabilities, which in turn promote post-sunset ionospheric irregularities. The occurrence and intensity of these irregularities varied spatially and temporally across the storms. For example, greater irregularity activity was observed in the Northern Hemisphere during equinoctial storms, while Southern

**Hemisphere dominance was seen during the solstice storm. In contrast, suppression of post-sunset irregularities was evident over the American sector during the intense storm in the winter season. Overall, the evolution and strength of post-sunset ionospheric irregularities at equatorial and low latitudes are governed by the combined influence of EIA density gradients, vertical plasma drifts, Local Time of maximum ring current development, and seasons.**

**These insights not only enhance our understanding of low-latitude ionospheric behavior but also provide valuable foundation for improving predictive models and mitigation strategies for satellite communication, navigation systems, and other technologies sensitive to ionospheric disturbances. Future research may focus on integrating multi-instrument observations with advanced model simulations to further unravel the complex electrodynamic processes governing storm-time ionospheric variability.**

Sincerely,

Dr. Nadia Imtiaz.

---

## Author Comment (AC2)

**General comments:**

This study presents the characteristics of equatorial and low latitude ionosphere during four intense geomagnetic storms that occurred on March 23-25, 2023, April 23-25, 2023, November 4-7, 2023 and May 10-13, 2024 in the ascending phase of Solar Cycle 25 based Joule Heating (JH) rates at high latitudes. For this purpose, authors have utilized the Weimer 2005 Model simulations to analyse the JH patterns along with Madrigal TEC maps to identify the changes in the intensity, location, and symmetry of the equatorial ionization anomaly (EIA) during these intense geomagnetic storms. Equatorial/low-latitude ionospheric plasma irregularities at different longitudes under geomagnetically disturbed conditions are studied using the TEC derived ROTI measurements. While a strong JH is observed during the May 2024 storm in the main phase, the other storms have the following order from strong to weak: March 2023, April 2023, and November 2023. The authors have investigated especially the role of asymmetric Joule heating in the structuring of the EIA like double crest, single crest or merged. Authors have utilized these features to investigate the formation or suppression of equatorial plasma irregularities. It is suggested that the generation of these ionospheric plasma irregularities and their latitudinal distribution strongly dependent on EIA's density gradients and latitudinal density structure. For instance, while the double crest EIA structures with strong plasma density gradients playing an important role in the generation of post sunset ionospheric plasma irregularities during the main phases of these four geomagnetic storms, single crest or merged EIA structure didn't favour the generation of ionospheric plasma irregularities. In addition, the role of storm-time penetration electric field in the structuring and seeding of ionospheric plasma irregularities have been investigated. Analysis of these storms by authors suggest that solar wind parameters, geomagnetic activity, JH, and PPEFs significantly influence ionospheric TEC variations, EIA crest formations, and postsunset plasma irregularities. These investigations revealed that these storms highlighted the importance of inter-hemispheric asymmetries in JH that affected the distribution and magnitude of ionospheric irregularities. Finally, the paper concludes that the resulting change in the thermospheric winds and electric fields due to storm conditions can alter the EIA structures which will impact the plasma irregularities at equatorial and low latitudes.

While many of the results presented here are already reported, I believe discussions like role of single EIA, double or Merged EIA on plasma irregularities looks to be interesting though it is known that asymmetry of EIA can suppress the plasma bubbles due to thermospheric meridional winds. I am providing the comments to the authors. The manuscript may be considered for publication only after implementing the comments made in this review report by incorporating appropriate changes.

**Specific comments**

It is mentioned that JH is strong in May 2024 Mother's Day storm during main phase. Then it followed like this: March 2023>April 2023>November 2023. Cause of variation proposed: Change in thermosphere winds and electric fields due to storm conditions alter EIA structures. What is the role of thermospheric neutral density such as O/N2 ratio on the EIA single crest, double crests?

Reply: Joule heating has a considerable and primarily indirect effect on the postsunset EIA during the main phase of a geomagnetic storms, which is mediated by storm-time neutral winds, thermospheric upwelling, and electric field modulation. Following to the Referee's advice, we have improved the Results and Discussion section.

**Whether authors have investigated positive/negative storms on the plasma irregularities.**

Reply: We agree with the referee suggestion. The present analysis primarily focused on storm-time electrodynamic drivers of plasma irregularities. The distinction between positive (density enhancement) and negative (density depletion) storm phases has not been explicitly separated in this study. Previous investigations have shown that positive storms can enhance background density and steepen the EIA crests, thereby favoring Rayleigh-Taylor growth, while negative storms tend to suppress irregularity development by reducing ambient plasma density. Incorporating such a classification in future case studies would strengthen the understanding of storm-phase-specific impacts on irregularity development.

**How the symmetry/asymmetry of EIA with single, double crests can influence the EPBs are not discussed. Please provide discussions.**

Reply: In the revised manuscript, we have added discussion on the structure of EIA and its influence on the EPBs. The combined influence of storm-time electric fields and thermospheric winds plays a key role in shaping the EIA, elevating plasma to higher altitudes, and redistributing electron density across trough and crest regions. During the main phase of storms, frequent southward/northward flipping in the interplanetary magnetic field (IMF Bz) can generate both eastward and westward prompt penetration electric fields (PPEFs), resulting from overshielding and undershielding effects. These alternating electric fields drive upward and downward vertical plasma drifts, further modifying EIA structures. In particular, EIAs with filled or deep troughs and steep vertical gradients in plasma density can create favorable conditions for the development of Rayleigh-Taylor instabilities, which in turn promote post-sunset ionospheric irregularities. The occurrence and intensity of these irregularities varied spatially and temporally across the storms. For example, post-sunset plasma irregularities are more likely to occur in the northern hemisphere during equinoctial storms, while southern hemisphere dominance was seen during the solstice storm. In contrast, suppression of post-sunset irregularities was evident over the American

sector during the intense storm in the winter season. Overall, the evolution and strength of post-sunset ionospheric irregularities at equatorial and low latitudes are governed by the combined influence of EIA density gradients, vertical plasma drifts, local time of the maximum ring current development, and seasons.

What is the role of electric fields on the generation/suppression of plasma irregularities? Reply: During geomagnetic storms, equatorial ionospheric plasma density distribution is influenced by electric fields, including DDEF from storm-time neutral winds and PPEF of magnetospheric origin astafyeva2018. According to (abdu1997, abdu2009), the presence of eastward/westward PPEF after sunset during the main phase of a storm can affect the vertical drift of plasma over the equatorial region either strengthening or disturbing the favorable conditions for plasma irregularities. During a storm's recovery phase, the westward DDEF in the sunset terminator may cause the plasma to move downward, and the normal PRE is reversed, reducing plasma irregularities (abdu1997).

Why authors have chosen TEC maps between 18:00-00:00 UT for all the storm events in Figure-6? Since different storms have main phases at different UT times, why authors have chosen this time interval to study the plasma irregularities and their connection to EIA.

Reply: We are interested to study the behavior of post-sunset ionospheric plasma irregularities during the main phase of different storms. For this reason we focused on time between 18:00-00:00 UT.

The plots shown in Figure-6 can be plotted as a line plots to show clear asymmetry in EIA. Also in the plot, map for 23:00 UT is not shown. Whether it is plotted like this only, please clarify.

Reply: We agree with the referee that line plots provide useful temporal variations at individual stations; however, TEC maps offer a significant advantage by showing the two-dimensional spatial distribution of the ionosphere. In particular, during storm-time conditions, TEC maps make it easier to visualize enhancements, and depletions across a wide region, providing a more comprehensive picture of the ionospheric response. We revised Figure 6 according to the referee suggestions with good colors and also included 23 UT.

Role of geomagnetic storms on plasma irregularities in the following papers: S. Sripathi and Ram Singh, A study on the response of the ionosphere to the three major space weather events of 2015 using meridional chain of ionosondes and GPS receivers over India, journal of Sun and Geosphere, DOI: 10.31401/SunGeo.2018.02.08, 2020. Similarly, the following paper discusses the role of geomagnetic storms and role of Es layers on the plasma irregularities. Please go through it. Singh, R., & Sripathi, S. (2020). A statistical study on the local time dependence of equatorial spread F (ESF) irregularities and their relation to low-latitude Es layers under geomagnetic storms.

Journal of Geophysical Research: Space Physics, 125, e2019JA027212. <a href="https://doi.org/10.1029/2019JA027212">https://doi.org/10.1029/2019JA027212</a>. It is said that the generation of ionospheric plasma irregularities and geographical distribution strongly dependent on EIA density gradients and general density structure. Please include the physics of their dependency.

Reply: Following paragraph has been added in the Results and Discussion section:

As reported in previous studies, both positive and negative storm phases have an impact on the latitudinal density gradients at the EIA crests and the background ionospheric density, both of which are essential for the growth of equatorial spread F (ESF) through the Rayleigh-Taylor instability Sripathi and Singh (2020); Singh and Sripathi (2020). These gradients are modified by storm-time electric fields such as PPEF and DDEF, which change the positions and intensities of ESF events. The ESF formation can also be further shaped by low-latitude sporadic E (Es) layers, which can locally increase densities and initiate localized irregularities Singh and Sripathi (2020). The observed dependency of plasma irregularities on the storm phase and the underlying ionospheric structures can be explained by these mechanisms.

It is mentioned that the storm-time Joule heating in high-latitude can cause atmospheric gravity waves (AGWs) to move equatorward, thus providing favorable conditions for formation equatorial/low-latitude ionospheric plasma irregularities. But it is under recovery phase mostly. But here authors are describing plasma irregularities in the main phase. Please clarify.

Reply: We agreed with referee that the storm-time Joule heating at high-latitudes can cause AGWs that can affect low-latitude ionosphere during the recovery phase of the storm. However, during intense geomagnetic storms, strong Joule heating during the main phase can also generate fast propagating traveling atmospheric disturbances (TADs). The combined effect of fast TADs atmospheric disturbances can also be seen in the form of EIA modification and ionospheric plasma irregularities during the main phase of the storms.

Double crest EIA structures with strong plasma density gradient play an important role in generation of post sunset plasma irregularities. Single crest or merged EIA does not favor the generation of irregularity. Please provide brief explanation for this. How the plasma irregularity generation is associated with EIA crest need to be mentioned clearly in the manuscript.

Reply: The necessary explanation has been added in the revised manuscript.

3.1 Structure of Equatorial Ionization Anomaly (EIA): "This merging and subsequent increase in TEC were driven by enhanced JH and geomagnetic activity". The claim can be elaborated by explaining the variation of JH with EIA structure.

Reply: Please see 3.1 section of the revised manuscript.

In general, PPEFs will affect the equatorial and low latitudes in the main phase of the storm, while Joule heating will affect the equatorial and low latitudes in the recovery phase of the storm through thermospheric wind circulation. However, here, authors are discussing the role of Joule heating during Main phase which I am not able to understand. Please clarify how the Joule Heating can impact the EIA crest in the main phase.

Reply: We agreed with the respected referee. The strong Joule heating during the main phase of the storm can affect the low-latitude ionosphere through thermospheric disturbances. Depending on the storm intensity and timing, these thermospheric disturbances along with PPEF lead to significant modification in EIA structure. We added relevant description in the mansucript.

The EIA trends are visually explained in table-3 with JH values where pattern can be observed. However, there is no mention of physics of their connection with the JH. What is the relation between JH and EIA dynamics are not clearly mentioned. Please explain. Similarly, Table-3 doesn't show any location where EIA crest is strong/weak. It has to be clearly mentioned.

Reply: Following discussion has been added: Joule heating has a considerable and primarily indirect effect on the post-sunset EIA during the main phase of geomagnetic storms, which is mediated by storm-time neutral winds, thermospheric upwelling, and electric field modulation. The strong JH causes an equatorward wind surge, which uses neutral drag to drive plasma along magnetic field lines to higher altitudes. This decreases the downward plasma diffusion and elevates ionospheric heights, reducing chemical loss and supporting positive ionospheric storm effects, which strengthen the EIA intensity. However, the EIA's structure typically undergoes the most significant changes in the evening, around the local sunset. After sunset, the equatorial ionosphere experiences PRE and vertical plasma drifts because of enormous Cowling conductivities and rapid changes in zonal winds. The combination of PPEF and PRE can significantly increase ambipolar diffusion and upward EXB drift, resulting in poleward extension of EIA crests. In contrast to poleward expansion, geomagnetically quiet or disturbed conditions may lead EIA crests to merge into a single peak across the geomagnetic equator. The electrodynamic effect, downward equatorial plasma drift, equatorward neutral winds, and an increase in the low-latitude O/N2 ratio all contribute to the structuring of EIA crests {balan2018, luan2021}. In the caption of Table 3, we have now added the location of EIA crests that is over the American longitudes.

3.2 Ionospheric Plasma Irregularities: The observations presented here under PPEFs and DDEFs and claims to have plasma irregularities. The proof/data/evidence/observation of the irregularities are needed to be included.

Reply: Following referee's suggestions, we have revised this section. Please see section 3.2 of the revised manuscript.

What is the role of atmospheric waves and tides on the variability of the PRE, EIA and plasma irregularities. It is known that equatorial electrodynamics plays significant role in day-to-day variability of the E X B drift and RT instability triggered plasma irregularities. So, this point needs to be addressed.

Reply: Following paragraph has been added:

Furthermore, the atmospheric waves and tides play a crucial role in the diurnal variability of the PRE, EIA, and the generation of plasma irregularities. Tidal and gravity wave perturbations originating in the lower atmosphere modify neutral winds, which in turn influence the equatorial dynamo and the vertical E × B drift Sripathi and Singh (2020). This variability alters the magnitude and timing of the PRE, modifies the EIA density gradients, and sustains the Rayleigh-Taylor instability, thereby controlling the occurrence and intensity of equatorial spread F (ESF) Singh and Sripathi (2020). Even under geomagnetically disturbed conditions, these processes highlight the importance of lower atmosphere-ionosphere coupling in regulating plasma irregularities.

Authors have chosen these four storms without looking into the role of seasons on the EIA. Usually, it is standard practice to choose storms that fall in equinox. Please address this point.

Reply: We thank to the referee for pointing this out. It should be noted that the four storms analyzed (March 23–25, 2023; April 23–25, 2023; November 4–7, 2023; and May 10–13, 2024) were selected primarily for their intensity and data availability rather than seasonal occurrence; while equinox storms are commonly emphasized in EIA studies, a systematic seasonal comparison is beyond the scope of this work and will be addressed in future investigations. Also, among the four storms examined, the May storm (near the June solstice) exhibited the most intense post-sunset Joule heating, followed by the March (equinox), April (post-equinox), and November (fall transition) storms, with the latter showing the weakest activity. Nearly symmetric heating patterns were observed during equinoctial storms, whereas solstice storms led to marked asymmetries between the two hemispheres.

If you want to study only 18:00-00:00 UT, please show this period with yellow highlight in the Figures-1-4 so that it is easy for the readers to follow it. Currently, authors have highlighted the main phase of the storm with 'red' rectangle box. In fact, in these figures, AE/AU/AL is missing.

Reply: In the revised manuscript, we have added the auroral indices and also highlighted the period of interest between 18:00-00:00 UT.

In the introduction, the sentence starts with 'lonospheric effects on applications are generally minor in mid-latitudes...' may be replaced with 'lonospheric effects are generally minor in mid-latitudes...' as the sentence is not conveying message correctly. Reply: Correction has been made.

"It is generally said that during storm time, it increases the daytime eastward electric field".  $E_{total}$  (t)= $E_{quiet}$  (t)+ $E_{PPEF}$  (t). It can either increase or decrease the equatorial zonal electric field, depending on IMF Bz polarity and whether you are on the dayside or nightside. This equation can be included.

Reply: We thank to the referee for suggesting this. Following text has been added in the revised version: During geomagnetic storms, the zonal electric field can be expressed as  $E(t)=E_Q(t)+E_{PPEF}(t)$ , where  $E_Q(t)$  is the quiet-time background field and  $E_{PPEF}(t)$  is the prompt-penetration electric field. Depending on the IMF  $B_z$  polarity and local time, the PPEF can either enhance the eastward daytime field or reduce (and sometimes reverse) the zonal field, leading to significant storm-time variability.

Correct 'Disturbed Dynamo (DD)' to 'Disturbance Dynamo (DD)'

Reply: Correction has been made.

---

## Author Comment (AC3)

**General comments:**

The manuscript is well-structured and addresses a timely and relevant topic in space weather and ionospheric physics. The use of multiple data sources enhances the reliability of the findings, and the interpretation is generally sound and supported by the data. The study contributes valuable insights into the behavior of ionospheric irregularities during geomagnetic storms in the early phase of Solar Cycle 25. The paper is suitable for publication in Annales Geophysicae after revisions. Below are some suggestions to improve clarity, scientific rigor, and presentation. Methodology and presentation:

The methodology is appropriate, but more detail on the ROTI calculation parameters (e.g. time resolution, thresholding) would improve reproducibility. Additionally, the criteria used to define the "intensity" of the selected storms beyond Dst values would worth being clarified.

Reply: ISEE ROTI data are derived from over 9300 GNSS phase observations acquired at 30-second intervals and transformed into 5-minute ROTI values at ionospheric pierce locations. The 2D ROTI maps are updated every 5 minutes with a  $0.25^{\circ} \times 0.25^{\circ}$  grid resolution. ROTI  $\geq 0.5$  TECU/min indicates ionospheric disturbances, while ROTI  $\geq 1.0$  TECU/min indicates significant irregularities.

We have used the SYM-H index rather than the disturbance storm time (Dst) index to characterize geomagnetic storms because SYM-H offers a 1-minute cadence, in contrast to the hourly resolution of Dst. While both indices represent the symmetric component of the ring current and thus share the same physical interpretation, the higher temporal resolution of SYM-H enables a more accurate depiction of the rapid variations and short-term dynamics of geomagnetic storms.

On the other hand, figures are generally clear and informative. Some color scales (e.g. in ROTI maps) could be better annotated for readability. Also please consider highlighting the storm periods in figures for better visual context. Figure 6 could include a 23:00 UT map or authors could clarify its absence.

Reply: Following referee's suggestion we have revised the Figures.

And Table 3 should be accompanied by a discussion of the physics behind the observed JH patterns and their relation to EIA dynamics.

Reply: Following discussion has been added as suggested by the referee: Joule heating has a considerable and primarily indirect effect on the post-sunset EIA during the main phase of a geomagnetic storms, which is mediated by storm-time neutral winds, thermospheric upwelling, and electric field modulation. The strong JH causes an equatorward wind surge, which uses neutral drag to drive plasma along magnetic field lines to higher altitudes. This decreases the downward plasma diffusion and elevates ionospheric heights, reducing chemical loss and supporting positive ionospheric storm effects, which strengthen the EIA intensity. However, the EIA's structure typically undergoes the most significant changes in the evening, around the local sunset. After sunset, the equatorial ionosphere experiences PRE and vertical plasma drifts due to enormous Cowling conductivities and rapid changes in zonal winds. The combination of PPEF and PRE can significantly increase ambipolar diffusion and upward E × B drift, resulting in poleward extension of EIA crests. In contrast to poleward expansion, geomagnetically quiet or disturbed conditions may lead EIA crests to merge into a single peak across the geomagnetic equator. The electrodynamic effect. downward equatorial plasma drift. equatorward neutral winds, and an increase in the low-latitude O/N2 ratio all contribute to the structuring of EIA crests (Balan et al., 2018; Luan, 2021).

Please explain how PPEF vs. DD electric fields interact with EIA structures and influence RT instability and PRE (pre-reversal enhancement).

Reply: Following discussion has been added as suggested by the referee: During geomagnetic storms, equatorial ionospheric plasma density distribution is influenced by electric fields, including DDEF

from storm-time neutral winds and PPEF of magnetospheric origin (Astafyeva et al., 2018). According to (Abdu, 1997; Abdu et al., 2009), the presence of eastward/westward PPEF after sunset during the main phase of a storm can affect the vertical drift of plasma over the equatorial region either strengthening or disturbing the favorable conditions for plasma irregularities. During a storm's recovery phase, the westward DDEF in the sunset terminator may cause the plasma to move downward, and the normal PRE is reversed, reducing plasma irregularities (Abdu, 1997).

**Results:**

The interpretation of EPB suppression during the April 2023 storm is compelling. It would be useful to compare this with similar events in previous solar cycles, if available and through this comparison, the selection of these specific storms could be justified. The discussion of hemispheric asymmetry is also interesting. Could the authors elaborate on possible seasonal or longitudinal effects that might contribute to this asymmetry? Please consider discussing the potential implications of these findings for GNSS signal degradation or navigation systems.

Reply: Following discussion has been added as suggested by the referee: The seasonal behavior of ionospheric plasma irregularities depends on magnetic declination (alignment of the Earth's magnetic field with the solar terminator) and trans-equatorial meridional winds Nigussie, 2016; Tsunoda, 1985). Lloyd's (Seba and includes winter (January, February, classification November. December), summer (May to August), and equinox (March, April, September, and October) (Liu et al., 2010). During equinotical months, the sun's greater elevation and minimal transequatorial thermospheric winds can result in significant ionization and steeper plasma gradients, which can sustain plasma irregularities (Rastogi, 1980; Maruyama, 1988). Intense plasma irregularities can significantly disturb GNSS-based navigation and communication systems by causing amplitude and phase scintillations. These scintillations cause fast fluctuations in signal power and carrier phase, which reduces positioning precision and reliability. During seasons conducive to the formation of plasma irregularities, notably around the equinoxes, GNSS receivers may encounter large range errors, cycle slipping, or even loss of lock, particularly for single-frequency users. As a result, understanding the seasonal and regional variability of irregularities is crucial for forecasting scintillation impacts and creating mitigation solutions, including the use of multi-frequency receivers, real-time monitoring, and augmentation systems (Aaron 1997, Basu et al., 1999).

**References:**

The references are appropriate and up to date. Consider adding a few recent studies on EPB dynamics during geomagnetic storms for broader context and also the following:

Tahir, A., Wu, F., Shah, M., et al. (2024) Multi-Instrument Observation of the lonospheric Irregularities and Disturbances during the 23-24 March 2023 Geomagnetic Storm Remote Sensing, 16(9), 1594. https://doi.org/10.3390/rs16091594. This study uses GNSS, Swarm, Formosat-7/COSMIC-2, and Fabry-Perot data to analyse EPB dynamics and electric field variations during the March 2023 storm, complementary perspective to the Zakharenkova, I., Astafyeva, E., et al. (2023) Development of the Storm-Induced Ionospheric Irregularities at Low and Mid-Latitudes e2021SW002891. **Space** Weather, 21(3), https://doi.org/10.1029/2021SW002891,

Focuses on the latitudinal expansion of irregularities during geomagnetic storms, relevant for comparing with the interhemispheric asymmetries discussed in the manuscript Imtiaz, N., Calabia, A., et al. (2025) Ionospheric Plasma Irregularities During Intense Geomagnetic Storms of Solar Cycle 25 EGUspherePreprinthttps://egusphere.copernicus.org/preprints/2025/egusphere-2025-86/.

The manuscript under review itself, for citation within future work or cross-referencing.

Reply: References have been added as suggested by the referee:

Language and style: The manuscript is generally well-written. A few minor grammatical issues and awkward phrasings should be corrected. For example,

- on page 2: "Storm-related variations in thermosphere/ionosphere parameters can dramatically"  $\rightarrow$  "Storm-related variations in thermosphere/ionosphere coupling/conditions can dramatically"

Reply: The rephrasing is incorporated according to the Referee's suggestion.

on page 2: "The EPF, the EIA, and the density gradients in the EIA should all be affected"  $\rightarrow$  "The EPF, the EIA, and the density gradients in the EIA are affected"

Reply: The rephrasing is incorporated according to the Referee's suggestion.

on page 2: "Disturbed Dynamo (DD)" → "Disturbance Dynamo (DD)".

Reply: The rephrasing is incorporated according to the Referee's suggestion.

on page 3: "Huang et al. (2013) studied that the single onset in the TEC"  $\rightarrow$  "Huang et al. (2013) showed that the single onset in the TEC"

Reply: The rephrasing is incorporated according to the Referee's suggestion.

- on page 5: "This article is organized as follows:"  $\rightarrow$  "This paper is organized as follows:" - on page 7: "In order to achieve this, we investigate four geomagnetic storms"  $\rightarrow$  "In order to gain these insights, we investigate four geomagnetic storms". Citation: https://doi.org/10.5194/egusphere-2025-86-RC3

Reply: The rephrasing is incorporated according to the Referee's suggestion.